# Offline Reinforcement Learning for Mixture-of-Expert Dialogue Management

**Dhawal Gupta**[*]
University of Massachusetts
dgupta@cs.umass.edu

**Yinlam Chow**
Google Research
yinlamchow@google.com

**Aza Tulepbergenov**
Google Research
atulep@google.com

**Mohammad Ghavamzadeh**[†]
Amazon
ghavamza@amazon.com

**Craig Boutilier**
Google Research
cboutilier@google.com

## Abstract

Reinforcement learning (RL) has shown great promise for developing agents for dialogue management (DM) that are non-myopic, conduct rich conversations, and maximize overall user satisfaction. Despite the advancements in RL and language models (LMs), employing RL to drive conversational chatbots still poses significant challenges. A primary issue stems from RL's dependency on online exploration for effective learning, a process that can be costly. Moreover, engaging in online interactions with humans during the training phase can raise safety concerns, as the LM can potentially generate unwanted outputs. This issue is exacerbated by the combinatorial action spaces facing these algorithms, as most LM agents generate responses at the word level. We develop various RL algorithms, specialized in dialogue planning, that leverage recent Mixture-of-Expert Language Models (MoE-LMs)—models that capture diverse semantics, generate utterances reflecting different intents, and are amenable for multi-turn DM. By exploiting the MoE-LM structure, our methods significantly reduce the size of the action space and improve the efficacy of RL-based DM. We evaluate our methods in open-domain dialogue to demonstrate their effectiveness with respect to the diversity of intent in generated utterances and overall DM performance.

## 1   Introduction

Natural language processing (NLP) has made significant strides in recent years, notably in the field of language generation. Thanks to advances in language modeling, particularly with the use of the transformer architecture (Vaswani et al., 2017), NLP models can now generate text that is often difficult to distinguish from that written by a human. However, despite these advancements, these models still fall short when it comes to having rich conversations. Current NLP models lack effective dialogue management: they are good at generating individual sentences, but struggle with maintaining coherent and engaging conversations. Whereas most compelling conversations generally span numerous topics, are rather open-ended, and often have an underlying goal (e.g., customer success, task completion, recommendation). This requires dialogue agents to understand the context of the conversation and respond appropriately while maintaining the ability to achieve goals.

*Reinforcement learning (RL)* is a natural approach to learning a policy for a dialogue management (DM) agent. Earlier work on RL-based dialogue systems relies on specific, hand-crafted semantic states (Levin and Pieraccini, 1997; Singh et al., 2002; Walker, 2000) or partially observable belief

---

[*]The work was done as a student researcher at Google Research (dhawgupta@google.com).
[†]The work was done prior to joining Amazon, while the author was at Google Research.

37th Conference on Neural Information Processing Systems (NeurIPS 2023).

states (Williams and Young, 2007; Young et al., 2010), in which the agent encodes conversations and chooses the best-structured dialogue action at each turn. Applications include relational reasoning (Shah et al., 2018), task completion (Shi and Yu, 2018), and query fulfillment (Serban et al., 2017), whose action spaces are structured enough to be represented by hand-crafted features. To handle more complex dialogues, recent approaches use language models (LMs) to extract semantic representations from conversation histories, treat them as dialogue states, and apply RL to learn a word-level generative DM agent (Jaques et al., 2019; Li et al., 2016, 2017; Shin et al., 2020).

However, unlike supervised learning approaches, where one can train imitation agents with offline conversation data, RL-based DM algorithms require online exploration. Unfortunately, constant interactions with real users are often expensive and time-consuming. While one can potentially address the DM problem using *offline* RL, issues such as model exploitation leading to distribution shift on the state and action spaces, when training on static datasets are of paramount concern (Levine et al., 2020). Moreover, the myriad variations of language make incorporating all possible conversation histories and bot utterances into the state and action spaces of an RL formulation of the DM problem impractical due to the combinatorics at play. As a result, naive application of RL to DM may result in poorly-performing agents that generate incomprehensible utterances (Zhao et al., 2019).

We tackle the above issues related to the use of offline RL in DM systems by leveraging recent advances in Mixture-of-Expert Language Models (MoE-LMs) (Chow et al., 2022). Specifically, we develop a suite of offline RL algorithms specialized in dialogue planning that exploit the structure of MoE-LMs. Our methods consist of three main components: **1)** a primitive LM, which uses a probabilistic encoder and decoder and is capable of generating diverse semantic intents; **2)** a number of *specialized* expert LMs, each of which generates utterances corresponding to a specific intent; and **3)** a compositional DM that, at each turn, given the encoded conversation history, selects an utterance from a set of candidate utterances suggested by the experts and pass it to the DM agent to execute.

Our contributions to offline RL adapted for MoE-based DM agents are four-fold. First, we exploit the hierarchical structure of MoE-LMs, allowing our offline RL methods to work with a significantly smaller, finite action space, making the RL problem more tractable. Second, by leveraging pre-trained MoE-LMs—which generate coherent utterances—and *regularization techniques* from offline RL that align the DM's behavior with that of the primitive LM—the proposed RL algorithms can focus on higher-level dialogue planning. The proposed combination results in higher data efficiency than standard RL methods by delegating the responsibility of language fluency to be handled by the MoE-LMs. Third, by using the diverse semantic representations of MoE-LMs, our methods operate at the sentence embedding space and have much simpler critic and actor updates. This circumvents the word-level credit-assignment issue, particularly challenging in long conversations (Saleh et al., 2020). Fourth, in contrast to the findings of Verma et al. (2022), where offline RL agents tend to lack utterance diversity (due to potential reward hacking and optimization of a single objective), our MoE-based DM agents by design are adept at generating utterances that reflect different intents.

We begin with a brief introduction of LMs, the MoE-LM architecture, and the use of Markov decision processes (MDPs) in DM in Section 2. We then describe the pre-training procedure for MoE-LMs—which encode diverse semantics and generate fluent utterances capturing specific intents—in Section 3. We derive state-of-the-art (SOTA) offline RL algorithms for training MoE-LMs in Section 4, and three MoE-LM specialized offline RL algorithms in Section 5. Finally, in Section 6, we evaluate our algorithms in open-domain dialogues against their ability to generate utterances with diverse intents and their overall DM performance.

## 2 Preliminaries

**Language Models (LMs)** In this work, we employ seq2seq LMs (Sutskever et al., 2014) to generate the next utterances in a dialogue. We assume access to a dataset of the form $\mathcal{D} = \{(\mathbf{X}^{(k)}, Y^{(k)})\}_{k=1}^{|\mathcal{D}|}$, where each $\mathbf{X}$ is an $L$-turn conversation history $\mathbf{X} = \{X_l\}_{l=0}^{L-1}$, wherein $X_l$ is the utterance in a conversation at turn $l$ and $Y$ is the next utterance. We define $N_{\mathbf{X}}$ to be an upper bound on the length (number of tokens) of each utterance $X_l$ in $\mathbf{X}$.[3] The role of a LM is to predict the probability of the next utterance $Y$, consisting of $N$ tokens, conditioned on the conversation history $\mathbf{X}$, i.e., $\Pr\left(Y = \{y_n\}_{n=1}^N \mid \mathbf{X}\right)$. In the transformer architecture (Wolf et al., 2019), a LM first encodes the conversation history $\mathbf{X}$ using an encoder $\Phi$ to a $(L \times N_{\mathbf{X}})$-length sequence of embeddings

---

[3]If $X_l$ has fewer tokens than $N_{\mathbf{X}}$, the remaining spaces will be padded by a specific token and masked.

$\{(z_{l,0}, \ldots, z_{l,N_{\mathbf{x}}-1})\}_{l=0}^{L-1}$, where each $z_{l,n}$ is a vector in the latent space induced by the encoder $\Phi$. For notational convenience, we concatenate these embeddings into a single embedding $z \in \mathcal{Z} \subseteq \mathbb{R}^d$, where $d$ is the overall dimension of the latent space. The next utterance $\widehat{Y} = \{\widehat{y}_n\}_{n=1}^N$ is then sampled, token-by-token, from a decoder $\Psi$, i.e., $\widehat{Y} \sim \Psi(\cdot \mid z) := \prod_{n=1}^N \Psi(\widehat{y}_n \mid \widehat{y}_0, \ldots, \widehat{y}_{n-1}; z)$, where $\widehat{y}_0$ is a fixed initial (start-of-sentence) token (Chien and Kuo, 2019) and the latent state is denoted as $z = \Phi(\mathbf{X})$.

**Markov Decision Processes (MDPs)** have been used to model dialogue management (DM) problems in a variety of settings (Li et al., 2016; Asadi and Williams, 2016; Jaques et al., 2019). In such MDPs, denoted by $\mathcal{M} = (\mathcal{S}, \mathcal{A}, P, r, s_0, \gamma)$, the state space $\mathcal{S}$ represents the tokenized conversation history and the initial state $s_0 \in \mathcal{S}$ is the initial user's query. The action space $\mathcal{A}$ is the tokenized language space with each action $a \in \mathcal{A}$ represents one possible next utterance of the agent. The transition kernel $P$ models the distribution over the user's response to the action taken by the agent (bot) and current conversational context. Finally, the reward function $r$ measures the user's satisfaction as a function of the conversation until the most recent step. In these MDPs, we can think of the LM as a policy that maps conversation histories to the next utterances. The goal is to find a policy $\pi^*$ with maximum expected discounted return, i.e., $\pi^* \in \arg\max_\pi J_\pi$, where $J_\pi := \mathbb{E}[\sum_{k=0}^\infty \gamma^t r_t \mid P, s_0, \pi]$. Note that the size of the tokenized state and action spaces grows exponentially with the vocabulary size. This makes it intractable to solve MDPs of this type, even for a medium-sized vocabulary.

**Mixture-of-expert Language Models (MoE-LMs).** Chow et al. (2022) recently demonstrated promising results using MoE-LMs to enrich a bot's utterances and improve DM (see Figure 1 for a sketch of the MoE-LM architecture). These results were achieved mainly due to (i) learning a language representation that captures different semantics (*primitive discovery*), (ii) a machinery, called *expert construction*, that embeds different intents into sub-models of this LM, so that they can behave appropriately when prompted, and (iii) a compositional dialogue manager module that comprehends the conversation and determines which response deems most appropriate.

For *primitive discovery*, one first learns a language model $\text{LM}_0 := (\Phi, \mathcal{G}_0, \Psi)$ that consists of a *stochastic encoder* $\mathcal{G}_0 \circ \Phi$, where $\Phi$ is an encoder mapping tokenized conversation histories $\mathbf{X}$ to a latent space $\mathcal{Z} \subseteq \mathbb{R}^d$ and $\mathcal{G}_0(z'|z) := \mathcal{N}(\mu_0(z), \sigma_0^2(z)\mathbf{I}_{d \times d})$ is a Gaussian distribution, and a decoder $\Psi$. $\text{LM}_0$ predicts the next utterance $\widehat{Y}_0$ (token-by-token) conditioned on $z'$ sampled from the latent distribution $\Psi(\widehat{Y}_0|z')$, i.e., $z' \sim \mathcal{G}_0(\cdot|z)$. We overload our notation to denote the *primitive* by $\text{LM}_0(Y|\mathbf{X}) := \mathbb{E}_{z' \sim \mathcal{G}_0(\cdot|z), z=\Phi(\mathbf{X})}[\Psi(Y|z')]$, which predicts the next utterance accurately and also has strong generalization in $\mathcal{Z}$ over a diverse set of possible utterances.

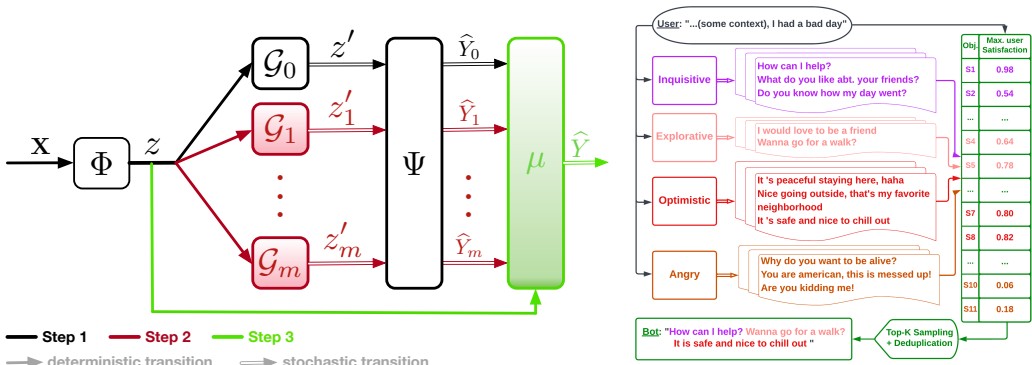

Figure 1: (Left) The MoE-LM architecture (Chow et al., 2022). Step 1: $\Phi$ encodes conversation history. Step 2: $\Psi \circ \mathcal{G}_i, \forall i$, generate candidate bot utterances. Step 3: The compositional dialogue manager $\mu$ selects the bot response by $Q$-score ranking and post-processing. (Right) Sample utterance workflow generated by a MoE-LM trained with Reddit data.

Given a primitive $\text{LM}_0$, the algorithm learns $m$ expert distributions $\{\mathcal{G}_i\}_{i=1}^m$, $\mathcal{G}_i(z'|z) = \mathcal{N}(\mu_i(z), \sigma_i^2(z)\mathbf{I}_{d \times d})$, each corresponds to a particular personality/intent and generates samples in specific parts of the latent space $\mathcal{Z}$. This results in $m$ LMs, $\{\text{LM}_i\}_{i=1}^m$, $\text{LM}_i := (\Phi, \mathcal{G}_i, \Psi)$, each serving as an *expert* that generates one or more candidate next utterances $\widehat{Y}_i$ that are relevant to the

conversation $\mathbf{X}$, and also compatible with its respective personality/intent. The compositional DM $\mu$ takes the encoded conversation history $z = \Phi(\mathbf{X})$ and candidate action utterances generated by the experts $\{\widehat{Y}_i\}_{i=0}^m$ as input, and selects one of them to execute, i.e., $Y \sim \mu(\cdot \mid z, \{\widehat{Y}_i\}_{i=0}^m)$. Given the state $s = \mathbf{X}$ and action $a = Y$, the MoE-LM policy that optimizes the DM MDP can be expressed as

$$\pi_{\text{MoE}}(a|s) = \int_{\{\hat{a}_i, z_i'\}_{i=0}^m} \mu\big(a|\Phi(s), \{\hat{a}_i\}_{i=0}^m\big) \prod_{i=0}^m d\Psi(\hat{a}_i|z_i') \, d\mathcal{G}_i(z_i'|\Phi(x)). \tag{1}$$

## 3  Warmstarting the Mixture-of-Expert LM

The MoE-LM approach effectively reformulates the reinforcement learning (RL) dialogue management problem, resulting in considerably smaller action spaces. This is achieved as the DM agent is now re now required to select the most appropriate utterance for presentation to the user from a finite, predefined set of candidate utterances, rather than generating responses word by word, hence allowing the agent to focus on optimizing the specific goal of the conversation task (as candidate utterances are separately optimized to follow particular bot-based characteristics/intents). Recall that the DM is a policy conditioned on both the latent state and the actions suggested by the experts. Before introducing the different RL methods for DM (Sections 4 and 5), in the following, we outline (i) the learning of diverse semantics (primitive LM) for conversation histories, which allows the agent to generate a wide variety of utterances, and (ii) the construction of specialized LMs (experts), which generate utterances with different intents.

Following the primitive discovery procedure in Chow et al. (2022), we learn the primitive LM, $\mathtt{LM}_0$, by solving the following KL-constrained optimization problem that aims at capturing diverse semantics:

$$\min_{(\Phi, \mathcal{G}_0, \Psi), \rho} \widehat{\mathbb{E}}_{z' \sim \rho(\cdot|z,Y), z=\Phi(\mathbf{X})}\big[-\log \Psi(Y|z')\big] \quad \text{s.t.} \quad \widehat{\mathbb{E}}_{z=\Phi(\mathbf{X})}\big[\text{KL}\big(\rho(z'|z,Y)||\mathcal{G}_0(z'|z)\big)\big] \leq \epsilon_{\text{KL}}. \tag{2}$$

In (2), $\widehat{\mathbb{E}}$ is the empirical expectation over $(\mathbf{X}, Y)$ in the dataset $\mathcal{D}$, $\rho$ is a distribution over the latent space conditioned on the encoded conversation history $z$ and the target utterance $Y$, and $\epsilon_{\text{KL}}$ is a positive real-valued threshold. The distribution $\rho(\cdot|z, Y)$ is a Gaussian $\mathcal{N}\big(\mu_\rho(z, \Phi_\rho(Y)), \sigma_\rho^2(z, \Phi_\rho(Y))\mathbf{I}_{d \times d}\big)$ in which $\Phi_\rho$ is a pre-trained encoder for the target utterance $Y$, and the mean $\mu_\rho(\cdot, \cdot)$ and variance $\sigma_\rho^2(\cdot, \cdot)$ are trainable models. Note that by solving (2), we maximize the log-likelihood of sentence $Y$ for a context and latent generation, while enforcing consistency between the latent variable $z'$ predicted by $\mathcal{G}_0(\cdot|z)$ and $\rho(\cdot|z, Y)$ via the KL constraint. In practice, we implement the KL constraint in (2) as a penalty weighted by a chosen coefficient.

To complete the MoE framework, one needs to train a set of experts $\{\mathtt{LM}_i\}_{i=1}^m$, each generating candidate utterances of different intents. By viewing each expert as a distribution of particular behaviors in conversation data $\mathcal{D}$, we leverage the results in Chow et al. (2022) and adopt a universal encoder-decoder $(\Phi, \Psi)$ among all the experts. In this view, each expert $i$ is a distribution $\mathcal{G}_i(\cdot|z)$ over certain regions of the latent space $\mathcal{Z}$. We train each $\mathcal{G}_i(\cdot|z)$ by solving

$$\min_{\mathcal{G}_i} \widehat{\mathbb{E}}_{z' \sim \mathcal{G}_i(\cdot|z), z=\Phi(\mathbf{X}), Y \sim \Psi(\cdot|z')}[-\ell_i(\mathbf{X}, Y)], \tag{3}$$

where $\ell_i(\mathbf{X}, Y) \in \mathbb{R}$ is a real-valued label that *characterizes* the intent of expert $i$. We can think of $\ell_i(\mathbf{X}, Y)$ as a score assigned to $Y$ resembling how strongly $Y$ exhibits the trait expert $i$ is meant to represent. Each expert is learned via *reward-maximization*, where $\ell_i$ is treated as an intent-aligned reward signal for expert $i$. Note that there is a connection between the above approach and contextual bandits (Chu et al., 2011), where both the context and action spaces are the latent space $\mathcal{Z}$, and the bandit policy is the latent distribution $\mathcal{G}_i$. The choice of greedy reward maximization is to encourage a particular behavior in the expert's immediate utterance rather than controlling its future utterances.

Long-term dialogue planning is handled by the compositional dialogue manager. For example, with Gaussian experts $\{\mathcal{G}_i\}_{i=1}^m$, we can use the standard REINFORCE algorithm (Sutton et al., 1999a), where the model parameters $(\mu_i, \sigma_i)$ are updated in the direction $\alpha \cdot \mathbb{E}_{z' \sim \mathcal{G}_i(\cdot|z), Y \sim \Psi(\cdot|z')}[\ell_i(\mathbf{X}, Y) \cdot \nabla_{\{\mu_i, \sigma_i\}} \log \mathbb{P}_{\mathcal{G}_i}(z'|z)]$ with learning rate $\alpha > 0$. To reduce the variance of these estimates, we can also adopt the baseline variance reduction technique (Greensmith et al., 2004).

# 4 RL for Mixture-of-Expert DM

In offline RL, the policy is learned from the collected conversations $\mathcal{D}$, without further online interactions. This potentially allows RL DM methods to leverage the abundance of offline conversational data for policy learning. Let $(\mathbf{X}, Y, X_+) \sim \mathcal{D}$ be a tuple sampled from the offline conversation data $\mathcal{D}$, with $X_+$ being the follow-up user response. We can formulate this tuple as a MDP data by defining $s := \mathbf{X}$, $a := Y$, $r(X_+)$, and $s_+ := (\mathbf{X}, Y, X_+)$ as the state, action, reward (w.r.t. the follow-up user response), and next state. A standard offline RL algorithm is $Q$-learning (Watkins and Dayan, 1992) that solves: $\min_Q \mathbb{E}_{(s,a,r,s_+)\sim\mathcal{D}}[(r + \gamma \max_{a_+} Q(s_+, a_+) - Q(s, a))^2]$.

However, with a large action space, the inner maximization (also termed as greedification) in $Q$-learning is generally computationally intractable. Furthermore, since one cannot ensure that the greedy $a_+$ is sampled from the same action distribution as in the offline RL dataset (an issue worsened by the large action set), such a covariate shift in the sampling distribution can cause an overestimation bias for the $Q$ estimate. To alleviate these issues, we propose to leverage the warm-started MoE-LM (Section 3), where the diverse semantic representation and the expert LMs are learned separately. This is crucial to make our offline RL DM problem tractable as the language fluency is captured by the MoE-LM, while our RL-based DM focuses on higher-level planning strategies. In the following, we describe how this can be achieved via different offline RL algorithms.

**Offline RL Methods for MoE-LMs:** One approach to address the aforementioned offline RL issues is *entropy regularization* (Haarnoja et al., 2018; Carta et al., 2021), which regularizes the greedification step to ensure the learned policy is either diverse enough or close to the behavior (data-generation) policy (e.g., through Shannon entropy or KL divergence between these policies). Recall that the primitive LM, $\text{LM}_0$, models the utterance distribution in $\mathcal{D}$ and the state-action-reward-next-state tuple $(s, a, r, s_+)$ of the DM MDP. With the following latent states generated by the primitive LM: $z = \Phi(s)$, $z_a = \Phi((s,a))$, and $z_+ = \Phi(s_+)$, we define the latent conversation data $\Phi(\mathcal{D})$ as a collection of $(z, z_a, r, z_+)$ tuples. With Shannon-entropy regularization, we can utilize the *soft actor critic* framework (Haarnoja et al., 2018) to develop RL updates for the *value function* $V(z)$, *state-action value function* $Q(z_a)$, and *latent generator* $\mathcal{G}(z'|z)$, which is initialized with the primitive latent expert $\mathcal{G}_0$. This framework minimizes the following losses:

$$L_Q = \mathbb{E}_{(z,z_a,r,z_+)\sim\Phi(\mathcal{D})}[(r + \gamma V_{\text{tar}}(z_+) - Q(z_a))^2] \tag{4}$$

$$L_V = \mathbb{E}_{z\sim\Phi(\mathcal{D}),(\hat{a},z')\sim\Psi\circ\mathcal{G}(.|z)}\left[Q_{\text{tar}}(z_{\hat{a}}) - \alpha \log \mathcal{G}(z'|z) - V(z)^2\right] \tag{5}$$

$$L_{\mathcal{G}} = \mathbb{E}_{z\sim\Phi(\mathcal{D}),(\hat{a},z')\sim\Psi\circ\mathcal{G}(.|z)}[Q(z_{\hat{a}}) - \alpha \log \mathcal{G}(z'|z)], \tag{6}$$

where the critic, $(V, Q)$, takes any encoded conversation history as input and predicts the corresponding cumulative return, $\alpha > 0$ is the entropy temperature, $(V_{\text{tar}}, Q_{\text{tar}})$ are the target value networks, $z' \sim \mathcal{G}(.|z)$ is the latent sample generated by $\mathcal{G}$, $\hat{a} \sim \Psi(z')$ is the utterance sampled from $\Psi \circ \mathcal{G}$, and finally $z_{\hat{a}} = \Phi((\mathbf{X}, \hat{a}))$ is the corresponding latent state.

From a *hierarchical RL* perspective (Sutton et al., 1999b; Saleh et al., 2020), the latent generator behaves like a high-level policy, whose latent sample $z'$ is used to generate a bot utterance via $\Psi$-decoding (with the primitive decoder $\Psi$ acting as the low-level policy). Extending the above RL updates to the case of relative-entropy (KL) regularization can be straightforwardly done by replacing the term $\log \mathcal{G}(z'|z)$ with $\log(\mathcal{G}(z'|z)/\mathcal{G}_0(z'|z))$, since the primitive LM approximates the behavior policy and the encoder-decoder pair $(\Phi, \Psi)$ is shared among the experts.

Multiple techniques in value-function parameterization have been employed to tackle the overestimation bias. Fujimoto et al. (2018) proposed maintaining two $Q$-functions, and a *dual $Q$-function* chooses the minimum value between them to avoid overestimation. Jaques et al. (2019) applies dropout in the $Q$-function to maintain an *ensemble* of $Q$-values, and outputs the minimum value to avoid overestimation. By utilizing these methods within the MoE-LM framework, we can propose the following variants of offline RL algorithms: (i) **SAC** that uses a dual $Q$-function and the actor-critic updates in (4)-(6), (ii) **EnsQ** that uses an ensemble of $Q$-functions and the same updates; and (iii) **KLC** that uses an ensemble of $Q$-functions and a latent KL-regularized actor-critic update.

Apart from the actor-critic approach that iteratively improves the value functions and policy, Implicit $Q$-Learning (IQL) (Kostrikov et al., 2021), a value-based offline RL algorithm, has recently shown success in tackling various problems, including task-oriented dialogue management (Snell et al., 2022). Within our MoE-LM framework, we propose the **IQL**-DM algorithm, whose value function $V(z)$ minimizes the following loss: $L_V = \mathbb{E}_{(z,z_a)\sim\Phi(\mathcal{D})}[L_2^\tau(Q_{\text{tar}}(z_a) - V(z))]$, where $L_2^\tau$ is the

expectile regression operator (Koenker and Hallock, 2001) of estimating the top-$\tau$ expectile statistics. The $Q$-function of IQL-DM is updated identically to that of actor-critic in (4), which estimates $Q(z_a) \approx r + \gamma V(z_+)$ via a least-square loss (Bradtke and Barto, 1996). The $V$-function estimates the top-$\tau$ quantile of the state-action $Q(z_a)$ random variable at every latent state $z$. As $\tau$ approaches one, $\tau \to 1$, the IQL updates converge to the optimal $Q$-function, $Q^*(z_a)$, i.e., $\mathbb{E}_{(z_a,r,z_+)\sim\Phi(\mathcal{D})}[(r + \gamma \max_{a_+} Q^*(z_{+,a_+}) - Q^*(z_a))^2] \to 0$, where $z_{+,a_+} = \Phi((\mathbf{X}, a, X_+, a_+))$ for any next-action utterance $a_+$. Intuitively, IQL leverages the generalization capacity of critic functions to estimate the value of the best action without directly querying the values of unseen actions. This makes it less conservative than most offline RL methods that constrain the policy's actions to be in-distribution via behavior regularization (e.g., **SAC**, **EnsQ**, **KLC**).

**Auto-regressive Decoding in Actor-Critic:** The actor-critic methods (SAC, EnsQ, KLC) ameliorated the two issues in offline RL to a certain extent (the inner maximization is replaced with $V$-function learning and covariate shift is controlled by policy entropy regularization). However, implementing these methods (Eqs. 5-6) entails sampling utterances from the current policy, i.e., $\hat{a} \sim \Psi \circ \mathcal{G}$, which involves expensive auto-regressive LM decoding at every training update. To resolve this issue, one may empirically replace $\Psi \circ \mathcal{G}$ with a *teacher-forcing* (Toomarian and Bahren, 1995) variant $\Psi_{\mathrm{TF}}(a) \circ \mathcal{G}$, which replaces auto-regressive decoding with a one-step generation from the bot utterance $a = Y$ in $\mathcal{D}$. This will further restrict the policy update of $\mathcal{G}$ to be close to the behavior policy. In contrast, since IQL does not perform explicit policy updates, it directly circumvents this expensive auto-regressive sampling operation of $\hat{a}$.

**DM Construction in MoE-LMs:** Recall that in an MoE-LM, the DM policy $\mu$ takes the encoded conversation history $z = \Phi(\mathbf{X})$, the $m + 1$ candidate action utterances generated by the experts $\{\widehat{Y}_i\}_{i=0}^m$, and selects one of them to execute, i.e., $a \sim \mu(\cdot \mid z, \{\widehat{Y}_i\}_{i=0}^m)$. Given the $Q$-function $Q(z_a)$ learned by any of the above offline RL algorithms, we extract the DM policy $\mu$ via softmax greedification over the finite set of MoE candidate utterances, i.e., $\mu(a \mid z, \{\widehat{Y}_i\}_{i=0}^m) \propto \exp(\beta \cdot Q(z_a))$, where $\beta > 0$ is the policy temperature. This DM policy uses the $Q$ function to score different candidate utterances and returns an utterance based on the likelihood of these scores.

# 5   Mixture-of-Expert Offline RL

In Section 4, we presented how state-of-the-art offline RL methods are adapted to the MoE framework, which can have limitations due to being agnostic to the model architecture. Recall that MoE dialogue management is a specialized hierarchical reinforcement learning (HRL) problem, which optimizes over a restricted class of DM policies defined by the convex hull of expert policy set (whose weights are defined by the DM policy $\mu$). This problem is of great interest because it reduces the original RL DM problem, with a combinatorial action space, into one that has a much smaller finite action set. In the following, we leverage the *mixture-of-policy* structure and develop offline RL algorithms that specifically target this HRL problem.

**Stochastic-action IQL (SAIQL):** The first approach applies IQL to the discrete, stochastic set of candidate action utterances $\{\widehat{Y}_i\}_{i=0}^m$ as generated by the MoE experts. Equipped with the latent conversation data $\Phi(D) = \{(z, z_a, r, z_+)\}$ (see Section 4) and the latent expert policies $\{\mathcal{G}_i\}_{i=0}^m$ in the MoE-LM, we propose a DM algorithm, whose value function $V(z)$ minimize the following loss:

$$L_V = \tfrac{1}{m+1} \sum_{i=0}^m \mathbb{E}_{z, \hat{a}_i \sim \Psi \circ \mathcal{G}_i(\cdot|z)} \big[ L_2^\tau \big( Q_{\mathrm{tar}}(z_{\hat{a}_i}) - V(z) \big) \big], \tag{7}$$

where $z_{\hat{a}_i} = \Phi((\mathbf{X}, \hat{a}_i))$ is the latent state that corresponds to the action utterance sampled from the $i$-th expert, $L_2^\tau$ is the expectile regression operator, and the $Q$-function is updated as in Eq. 4. To incorporate the maximization over candidate utterances in IQL, we compute the expectile regression over the joint latent state and expert policy distributions.

However, unlike the standard IQL DM algorithm, which avoids autoregressive decoding for policy execution, SAIQL requires auto-regressive sampling of all $m + 1$ candidate utterances. Suppose the augmented latent conversation data $\Phi(D)_{\mathrm{SA}} = \{(z, z_a, r, z_+, \{z_{\widehat{Y}_i}\}_{i=0}^m)\}$, which also includes the set of latent expert actions $\{z_{\widehat{Y}_i}\}_{i=0}^m$, is available. One straightforward way to circumvent this issue is by replacing the expectation over experts with the realized candidate utter-

ances, i.e., by approximating the value function in SAIQL with its unbiased empirical average $\frac{1}{m+1} \sum_{i=0}^m \mathbb{E}_{(z,\{z_{\widehat{Y}_i}\}_{i=0}^m) \sim \Phi(\mathcal{D})_{\text{SA}}}[L_2^\tau(Q_{\text{tar}}(z_{\widehat{Y}_i}) - V(z))]$.

While having access to candidate utterances is not standard in IQL, it is necessary here to allow $Q$-learning to exploit quantile regression over *realized* candidate utterances (an approach shown to be sound in Boutilier et al. (2018)). Therefore, we termed this method *stochastic action* IQL (**SAIQL**) to reflect the stochastic action sets used in IQL training. Once **SAIQL** converges, the DM policy is also constructed as a softmax of $Q$-values applied to each candidate utterance.

The **MoE-MDP** is defined as $\bar{\mathcal{M}} = (\bar{\mathcal{S}}, \bar{\mathcal{A}}, \bar{P}, \bar{r}, \bar{s}_0, \gamma)$, where the state space is the product of the learned latent space $\mathcal{Z}$ and the joint action space of the $m + 1$ experts, i.e., $\bar{\mathcal{S}} = \mathcal{Z} \times \mathcal{A}^{m+1}$; the action space consists of the $m + 1$ experts, i.e., $\bar{\mathcal{A}} = \{0, \ldots, m\}$; the initial state $\bar{s}_0$ is the encoding of the initial user's query and the utterances suggested by the experts in response to this query; the transition kernel models both the user's responses and the next experts' actions; and finally the reward is the same as in the original MDP. Since MoE-MDP has a finite number of actions, learning a policy $\lambda$ is equivalent to solving a finite-action MDP, i.e., $\lambda^* \in \arg\max_\lambda J_\lambda := \mathbb{E}[\sum_{k=0}^\infty \gamma^t \bar{r}_t \mid \bar{P}, \bar{s}_0, \lambda]$.

**Follow-the-Leading-Expert (FtLE):** Banijamali et al. (2019) showed that the MoE-MDP problem is NP-hard but can be approximated by $\max_{\lambda \in \Delta^{m+1}} \sum_{i=0}^m \lambda(i) V^i(z) + \mathcal{U}(\bar{\mathcal{M}})$, where $V^i$ is the value function of the $i^{\text{th}}$ expert and $\mathcal{U}(\bar{\mathcal{M}}) > 0$ is a surrogate function that depends on the experts' stationary distributions. However, computing these distributions is generally intractable as the experts are LMs themselves. This motivates our heuristic **FtLE** algorithm, which ignores the second term in training a set of expert critic functions and picks the best action at each step. To efficiently parameterize the critic function, similarly to the architecture used in DQN (Mnih et al., 2013) for discrete-action RL, we define a $(m + 1)$-headed critic function, where each head represents the value of following an expert's policy. To train the multi-headed critic, we modify the standard critic losses as

$$L_Q = \sum_{i=0}^m \mathbb{E}_{z,z_a,r,z_+}\left[(r + \gamma V_{\text{tar}}^i(z_+) - Q^i(z_a))^2\right], \quad L_V = \sum_{i=0}^m \mathbb{E}_{z,\hat{a}_i \sim \Psi \circ \mathcal{G}_i(\cdot|z)}\left[(Q_{\text{tar}}^i(z_{\hat{a}_i}) - V^i(z))^2\right], \quad (8)$$

where $Q^i$ and $V^i$ represent the critic-function head for expert $i$. To overcome the auto-regressive sampling issue in (8), we relabel the offline conversation data $\mathcal{D}$ by assigning action utterances to train the critic function(s) whose corresponding expert(s) most likely generate those utterances. Specifically, consider the following V-function loss:

$$L_V = \sum_{i=0}^m \mathbb{E}_{z,z_a,Y}\left[\mathbf{1}_{i=i(z,Y)} \cdot \left(Q_{\text{tar}}^i(z_a) - V^i(z)\right)^2\right], \quad (9)$$

where $\mathbf{1}_{i=i(z,z_a)}$ selects the expert based on the best log-likelihood $i(z, Y) := \arg\max_i \log \Psi(Y|z^{i,\prime})$ with $z^{i,\prime} \sim \mathcal{G}_i(\cdot|z)$. After learning the critic functions, the **FtLE** DM policy can be constructed as $\mu(a \mid z, \{\widehat{Y}_i\}_{i=0}^m) \propto \exp(\beta Q^{i(z,a)}(z_a))$.

**Value-based RL for MoE-MDP (MoE-VRL)**: Consider a $(m + 1)$-headed value function $\Lambda$ of the MoE-MDP, where each head represents the optimal value by choosing the corresponding expert's action. Applying standard DQN, this function can be learned by minimizing the following loss:

$$L_\Lambda = \mathbb{E}_{z,Y,r,z_+}\left[\left(r + \gamma \max_{i_+} \Lambda_{\text{tar}}(z_+, i_+) - \Lambda(z, i(z, Y))\right)^2\right], \quad (10)$$

where $\Lambda_{\text{tar}}$ is the target $\Lambda$-network. For simpler exposition, we only use the partial MoE-MDP states of encoded conversations in the above DQN loss and omit the candidate action utterances. Extending to the full MoE-MDP state is straightforward but is omitted for brevity. The inner maximization over $i_+$ can be computed explicitly because the MoE-MDP action space of expert indices is finite and small. Here, $i(z, Y)$ is the same index function that attributes utterance $Y$ to the expert most likely to generate it (based on likelihood). With the optimal value function $\Lambda^*(z, i)$, the MoE-MDP policy picks the best expert $\lambda^*(z) := \arg\max_i \Lambda^*(z, i)$, and the DM policy can be constructed as $\mu(a \mid z, \{\widehat{Y}_i\}_{i=0}^m) \propto \exp(\beta Q^{\lambda^*(z)}(z_a))$, where $Q^{\lambda^*(z)}(z_a)$ is the critic of the optimal expert.

## 6 MoE-based DM Experiments

We evaluate our MoE-based offline RL algorithms on two open-domain benchmarks common in the RL-based dialogue management literature (Jaques et al., 2019). The first one is the Cornell Movie

corpus (Danescu-Niculescu-Mizil and Lee, 2011), which consists of conversations between speakers in different movies. The second is the Reddit Casual (Ghandeharioun et al., 2019) conversations dataset, which is a subset of the Reddit corpus that only contains casual conversations.

**Environment:** We perform the experiment by having DM agents interact with a DialoGPT (Zhang et al., 2019) simulated-user environment. The task is to maximize user satisfaction, which is measured by the user's overall sentiment. To construct an immediate reward, we set $r(X_+) := \ell_{\text{sent}}(X_+)$, where $\ell_{\text{sent}}(X)$ is a RoBERTa-based sentiment classifier (Liao et al., 2021), which assigns a score from $[-1, 1]$ that is inversely proportional to the (negative) positive sentiment prediction probabilities.

We pre-train the MoE-LM with either the Cornell or Reddit dataset and construct 10 experts (i.e., $m = 9$, plus the primitive expert), each corresponding to an individual intent in open-ended dialogues, including "empathy", "optimism", "cheerfulness", "contentment", "dejection", "rage", "sorrow", "questioning", "exploration", etc. See Appendix B for details. The conversation lasts for 5 turns (with $\gamma = 0.8$), where each turn entails a query/response from the user followed by an agent's utterance. During the agent's turn, each expert generates 5 candidate utterances, thus resulting in a total of 50 candidate utterances. To evaluate the methods, we measure the return of the trajectory generated by different agents via $\mathbb{E}_{\mathbf{X}_0 \sim \mathcal{D}}[\sum_{i=0}^{4} \gamma^i r(X_{i+1}) | Y_i \sim \text{LM}(.|\mathbf{X}_i), X_{i+1} \sim P_{\text{Dialog-GPT}}(.|\mathbf{X}_i, Y_i)]$.

**Evaluation:** We employ two evaluation approaches, namely (i) a model-free approach that only utilizes the learned $Q$ function to score candidate utterances, and where the DM policy selects the action utterance based on a softmax likelihood; and (ii) a model-based approach that uses the Value function ($V$) along with a learned next-user utterance model $P_{\text{user}}(X_+|z_Y)$, that optimizes the following loss: $L_{P_{\text{user}}} = \mathbb{E}_{(z_a, r) \sim \mathcal{D}, \hat{X}_+ \sim P_{\text{user}}(.|z_a)}[(r - r(\hat{X}_+))^2]$. We first approximate the $Q$ function via $Q(z_a) \approx r(\hat{X}_+) + \gamma V(\hat{z}_+)$, where $\hat{X}_+$ denotes the next user utterance sampled from $P_{\text{user}}(.|z_a)$, then use that function to score candidate utterances, and, finally have the DM policy select the action utterance analogously. Human evaluation is also conducted on the DM performances of different offline RL agents. More details and results can be found in Appendix D.

Table 1: SOTA offline RL methods

| Method | Reddit Casual | | Cornell | |
|---|---|---|---|---|
| | Model Free | Model Based | Model Free | Model Based |
| IQL | $0.53 \pm 0.47$ | $\mathbf{4.25 \pm 0.12}$ | $-1.32 \pm 0.19$ | $\mathbf{1.47 \pm 0.15}$ |
| SAC | $\mathbf{0.97 \pm 0.52}$ | $4.13 \pm 0.21$ | $-1.55 \pm 0.19$ | $0.36 \pm 0.26$ |
| EnsQ | $0.10 \pm 0.40$ | $4.06 \pm 0.25$ | $-1.51 \pm 0.20$ | $0.21 \pm 0.21$ |
| KLC | $0.31 \pm 0.46$ | $3.69 \pm 0.37$ | $-1.46 \pm 0.21$ | $-0.07 \pm 0.25$ |
| BC | $-0.65 \pm 0.41$ | | $-2.18 \pm 0.36$ | |
| Bandits | $\mathbf{4.3 \pm 0.16}$ | | $1.3 \pm 0.17$ | |

Table 2: MoE specific offline RL methods

| Method | Reddit Casual | | Cornell | |
|---|---|---|---|---|
| | Model Free | Model Based | Model Free | Model Based |
| EXP 1* | $0.97 \pm 0.52$ | $4.25 \pm 0.12$ | $-1.32 \pm 0.19$ | $1.47 \pm 0.15$ |
| SAIQL | $0.81 \pm 0.42$ | $\mathbf{4.65 \pm 0.06}$ | $-1.34 \pm 0.25$ | $2.61 \pm 0.24$ |
| FtLE | $\mathbf{1.14 \pm 0.49}$ | $4.59 \pm 0.07$ | $\mathbf{-0.39 \pm 0.24}$ | $3.51 \pm 0.19$ |
| MoE-VRL | $0.72 \pm 0.47$ | $4.46 \pm 0.10$ | $-0.58 \pm 0.24$ | $\mathbf{3.62 \pm 0.17}$ |

**Experiment 1: SOTA Offline RL with MoE-LMs:** The goal of this experiment is to investigate the effectiveness of SOTA offline RL algorithms. In these experiments, we only make use of the primitive language model $\text{LM}_0 = (\Phi, \mathcal{G}_0, \Psi)$ to generate sample utterances. To simulate previous works using single policy settings, we fine-tune the latent base distribution $\mathcal{G}_0$ for policy optimization while keeping the encoder-decoder $(\Phi, \Psi)$ fixed. As mentioned in Sec. 4 we deploy the following offline RL algorithms to train the DM policy $\mu$ of MoE-LMs: (i) **SAC** (Haarnoja et al., 2018) with a dual $Q$ function critic (Fujimoto et al., 2018); (ii) **EnsQ**, which utilizes an ensemble of $Q$ functions (Jaques et al., 2019) with actor-critic; (iii) **KLC** (Saleh et al., 2020), which utilizes the dual $Q$ function and applies KL regularization between the latent policy $\mathcal{G}$ and the primitive policy $\mathcal{G}_0$,

i.e., $\mathbb{E}_{\mathcal{G}(\cdot|z)}[\log(\mathcal{G}(z'|z)/\mathcal{G}_0(z'|z))]$ in the actor-critic algorithm update [4]; (iv) **IQL** (Kostrikov et al., 2021), which adopts the idea from Q learning to estimate an optimal $Q$ function in the MoE-LM latent space. To our knowledge, our work is among the first that makes use of IQL for open-domain dialogue management. These methods have been implemented in ways where the original idea has been preserved, making the comparison fair to the original works. With each learned $Q$ function, the bot picks the final action by sampling from a softmax distribution of $Q$ scores overall candidate utterances. To demonstrate the efficacy of offline RL methods, we also include results from Behavior Cloning (**BC**) as well as simple reward maximization (**Bandit**) (i.e., $\gamma = 0$) for comparisons.

Table 1 presents the results of our experiments with these methods in the open-dialogue system, where a 5-turn conversation was generated. The table displays the mean return over 100 conversations with their respective standard errors. Our experiments demonstrate that model-based evaluation can significantly improve dialogue management over the model-free counterpart, even with a next-user LM that is much simpler than the Dialog-GPT user. Among most model-based and model-free evaluations, we found that **IQL**, originally designed to tackle offline RL problems, outperforms other RL methods. This performance can be attributed to IQL's ability to (i) alleviate $Q$ overestimation errors due to co-variate shifts; (ii) estimate the optimal values without being overly conservative w.r.t. the behavior policy, and (iii) avert the auto-regressive utterance sampling issues in training.

Interestingly, we also found that **KLC** and **EnsQ**, two standard methods in RL-based DM, struggled to achieve satisfactory performance in our experiments. This may be due to the fact that applying dropout (for ensemble $Q$) and KL regularization in the fixed MoE-LM latent space makes DM algorithms overly conservative. In contrast, **SAC** successfully learns a well-performing model-free DM policy but fails in the model-based regime, potentially demonstrating its instability in critic-function learning. **BC** also fails to provide any satisfactory performance on any of the domains, and surprisingly, **Bandit** method or plain reward maximization did as well as **IQL**, pointing to the fact that maybe the offline RL methods being used or not exactly helping in planning at all.

**Experiment 2: MoE-specific Offline RL:** In this experiment, we explore the benefits of leveraging the MoE framework for training offline RL agents in open-domain conversational systems. Building upon the insights from our previous experiment (Experiment 1), we propose several modifications to standard Offline RL algorithms to take advantage of the MoE framework. As mentioned in Sec. 5, we developed the following MoE-specific offline RL algorithms for DM: (i) **SAIQL**, which extends IQL to incorporate the multiple candidate utterances generated by the experts; (ii) **FtLE**, which learns a DM policy to follow the best expert policy at each step (estimation of the experts' long-term values is done concurrently with a multi-headed critic architecture and data relabeling) and (iii) **MoE-VRL**, which learns an optimal meta-value function over the space of experts. Leveraging the MoE-MDP formulation, solving which leads to an optimal DM policy that provides the optimal sequences of expert policy switching. We aim to evaluate the potential of these MoE-specialized offline RL algorithms over off-the-shelf offline RL methods in DM.

Table 2 shows the return observed similar to the ones displayed in Table 1. The first row in the table displays the best performance across all methods from Experiment 1, for comparison. Our results demonstrate the efficacy of the proposed methods that utilize the structure of the MoE framework in dialogue management. All the methods that used all experts while training (**SAIQL**, **FtLE**, and **MoE-VRL**) outperformed the SOTA offline RL methods, indicating that an offline RL algorithm that takes the candidate utterances into account can generally improve dialogue planning. Moreover, making the RL algorithms attuned to the multiple-expert structure (i.e., **FtLE** and **MoE-VRL**) indeed results in even better DM performance, emphasizing the benefits of reformulating the DM MDP using the HRL paradigm, where the DM policy is optimized over a restricted class of finite-action policies. Also, we note that only MoE-aware offline RL methods were actually able to outperform simple per-step greedification (i.e. **Bandit**) which hints to the fact that they were actually able to plan ahead and perform long-term credit assignments to optimize return. Whereas all the standard offline RL methods failed to do that (Table 1). Using multiple critic functions to separately estimate the value of different experts also allows us to better understand their long-term utility (of the corresponding intents) and how they affect the conversation quality. Overall, these findings highlight the potential of the MoE-specific offline RL methods to improve dialogue management performance.

**Human Evaluation:** Table 3 summarize the results of around 600 ratings provided by 80 on the bots' quality, in terms of fluency, and conversation-level sentiment improvement on the Reddit Casual

---

[4]The RL DM approach in Jaques et al. (2019) which applies KL regularization at the word-level LM policy is not applicable to our case because our DM policy is defined in the latent space.

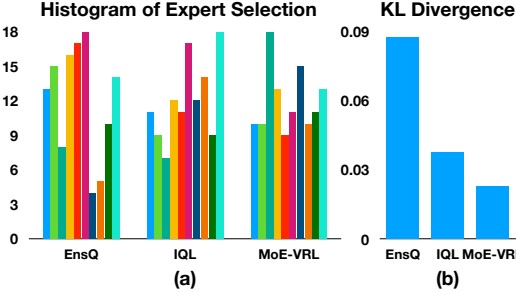

Figure 2: Experiment on the Cornell dataset with Model-based evaluation(a) Histogram of frequency of expert selection. (b) KL divergence against uniform distribution.

Table 3: Human raters evaluation

| Method | Avg. Fluency | Sentiment |
|--------|--------------|-----------|
| BC | $0.67 \pm 0.26$ | $0.24 \pm 0.50$ |
| KLC | $0.62 \pm 0.27$ | $0.66 \pm 0.47$ |
| IQL | $0.84 \pm 0.24$ | $0.72 \pm 0.46$ |
| SAIQL | $0.81 \pm 0.19$ | $0.57 \pm 0.50$ |
| FtLE | $\mathbf{0.88 \pm 0.24}$ | $\mathbf{0.76 \pm 0.48}$ |
| MoE-VRL | $0.72 \pm 0.28$ | $0.70 \pm 0.45$ |

dataset. We can similarly notice in this case that MoE specific offline RL methods seems to perform better both in terms of fluency and sentiment improvement over standard offline RL. The complete details of the experiment can be found in Appendix D.

**Experiment 3:** In this experiment, one aims to investigate the effectiveness of selecting different experts during dialogue management. To this end, we conduct a study where we measure the frequency with which utterances from different experts are selected throughout the conversation. Specifically, we wish to understand the diversity of intents selected by different offline RL algorithms.

Given approximately 200 conversation turns, we measure the frequency of the expert agents when their utterances are selected and present the frequency metric for the worst performing offline RL method (**EnsQ**), a good performing method (**IQL**), and an MoE-specific RL algorithm (such as **MoE-VRL**). To visualize our findings, we plot a histogram of the frequencies of different experts being selected and calculate the KL divergence of the histogram against a uniform distribution over the experts. While the uniform distribution may not be the optimal distribution of utterances, it provides a measure of how well the agents make use of different experts, along with their performance.

Figure 2 illustrates the result for the above experiment for the Cornell dataset, in the model based setting. We observe that the worst performing agent, **EnsQ**, has a highly skewed distribution of expert selections, with a few experts being heavily favored over others. This suggests that **EnsQ** is less diverse and does not effectively utilize the full range of expert capabilities available. On the other hand, both **IQL** and **MoE-VRL** exhibit a more balanced distribution of expert selection, with utterances chosen from multiple experts throughout the conversation; i.e., their frequency distributions are closer to a uniform distribution, with much lower KL divergence distance.

However, note that there is a clear performance gap between **MoE-VRL** and **IQL** where former significantly outperforming the latter. This highlights the importance of incorporating the MoE framework to better utilize the intent of different experts in dialogue planning, rather than relying on generating a diverse set of candidate utterances. Overall, these results suggest that encouraging diversity in intents and better utilizing expert knowledge in planning can improve DM performance.

## 7 Concluding Remarks

By leveraging the recent advances of MoE-LMs, we developed a suite of offline RL-based DM algorithms. Our methods significantly reduce the action space and improve the efficacy of DM. To understand how well our offline RL approaches generate diverse utterances and solve DM problems, we evaluated them on two open-domain dialogue tasks and compared them with SOTA offline RL baselines. Our results showed that by exploiting the MoE-LM structure, our specialized offline RL DM methods (i) improve the diversity of intents in bot utterances; (ii) have better sample efficiency; and (iii) yield better overall performance in both the model-based and model-free settings. Our work provides important insights on how to create scalable RL-based DM methods that train chatbots to achieve dialogue tasks and enhance user satisfaction. Future work includes fine-tuning the experts (i.e., low-level policies) with offline RL, learning the optimal semantic representation for hierarchical RL, preventing dialogue agents from generating harmful behaviors (e.g., by enforcing safety constraints in the RL algorithms), and evaluating our DM methods on more realistic problems, such as customer support, conversational recommendation, and persuasion.

## Acknowledgement and Funding Disclosures

We express our sincere gratitude to the Mudcats team at Google Research for their invaluable feedback and innovative ideas, which have significantly enriched this project. We would also like to thank the anonymous reviewers, whose insightful suggestions and constructive feedback were instrumental in enhancing the clarity and overall presentation of the paper. We acknowledge the generous funding and computational resources provided by Google Research, crucial for the completion of our work.

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

# A  Additional Results

## A.1  Diversity over all agents and Datasets

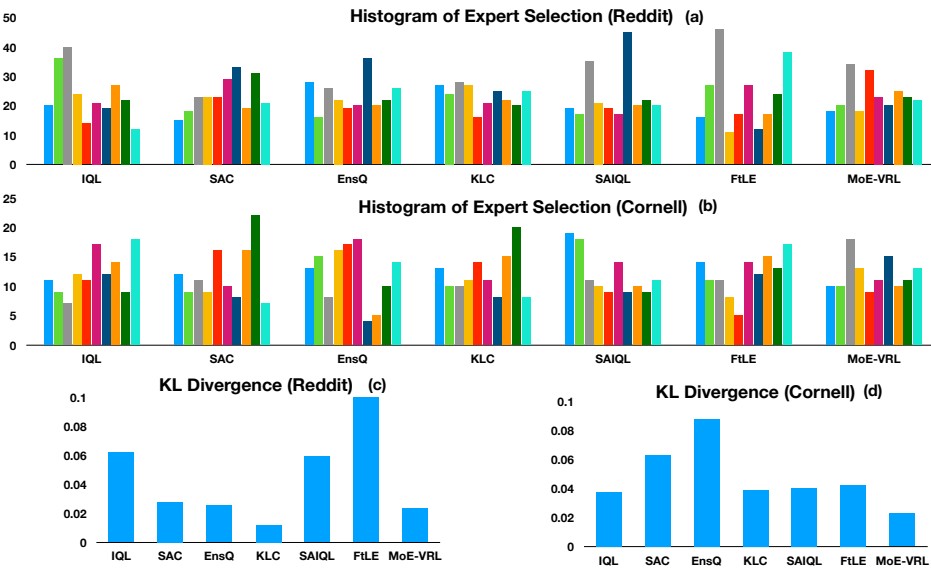

Figure 3: Diversity for all agents (a) Reddit with Model-based approximation, (b) Cornell with Model-based approximation, (c) and (d) depict the KL divergence of all agents w.r.t. to uniform distribution for Reddit and Cornell.

## A.2  Different Metrics for MoE-LM's

To measure the quality of LMs learned in MoE-LM we measure the following three metrics, similar to Chow et al. (2022) for 25 generated utterances. **Diversity** : measured as 1 - Sparsity Hurley and Rickard (2009) of the singular values of the embedded utterances, **Gram- {1,2,3}** Li et al. (2015) : Ratio of unique {uni, bi, tri}-gram in generated utterances, and finally **Perplexity** Bahl et al. (1983).

Table 4: Diversity, Gram-{1,2,3} and Perplexity of the primitive LM on Reddit casual and Cornell

| Dataset | Diversity | Gram-1 | Gram-2 | Gram-3 | Perplexity |
|---------|-----------|--------|--------|--------|------------|
| Reddit | $0.14 \pm 0.05$ | 0.35 | 0.77 | 0.90 | $38.81 \pm 17.34$ |
| Cornell | $0.12 \pm 0.04$ | 0.31 | 0.60 | 0.79 | $43.87 \pm 28.81$ |

Table 5: Quality of each expert trained on Reddit casual & Cornell with respect to their trained label.

| Dataset | Question | Exploration | Positive Sent. | Negative Sent. | Sent. Coherence | Joy | Optimism | Anger | Sadness |
|---------|----------|-------------|----------------|----------------|------------------|-----|----------|-------|---------|
| Reddit | $0.95 \pm 0.27$ | $0.47 \pm 0.21$ | $3.29 \pm 0.33$ | $1.42 \pm 0.38$ | $0.51 \pm 0.40$ | $1.99 \pm 0.38$ | $1.25 \pm 0.43$ | $1.48 \pm 0.39$ | $2.01 \pm 0.46$ |
| Cornell | $1.58 \pm 0.39$ | $0.33 \pm 0.17$ | $3.55 \pm 0.99$ | $1.90 \pm 0.5$ | $0.69 \pm 0.40$ | $2.44 \pm 0.71$ | $2.11 \pm 0.99$ | $2.71 \pm 0.69$ | $3.45 \pm 0.83$ |

## A.3  Sample Utterances for Reddit Chitchat

### A.3.1  MoE-VRL

1. <conv history> user: 11:15 here, it's been a day for sure. agent: I am so tired. user: For you it has been a day but in your case I am not so sure. agent: how are you doing?

2. <conv history> user: I have my test tomorrow morning, second try. agent: Failed first try because on a parallel park I was too far away from the curb. user: Hmm my driving test didn't have any parking in it. agent: Source: took my test a couple of months ago user: In the UK you have to do a maneuver and the one my tester gave me was parallel park.

3. <conv history> user: I'm getting back into writing and reading to be more immersed in the concept of writers. agent: It's always a good distraction for things like this. user: Being able to use your phone is crucial. agent: I can't wait to dive into something more physical though it's been a while since I felt distracted by internet. user: Good Good points! agent: I'm glad to hear the science behind writing has been very active. user: The past few days I was reading about some writers in America who seemed very much to embrace the science of writing! agent: It's really refreshing to see another view from another perspective. user: There's nothing I really care about as a person. agent: I'm glad to hear the internet is bringing so much joy to people of needle, and it's really encouraging to read stories about other people doing the same thing to make the internet more loving and sustain the quality of the medium.

4. <conv history> user: Yeah I feel you. agent: 28 here never been in a relationship. user: I've asked dudes out and it always ends with "sorry, not attracted to black girls" agent: I'm considering going out with one of those user: If an encounter deserves to be raped, then your defense would be ridiculous. agent: If a lady wants to cheat on you, then so be it. user: Years of my life in general have been destroyed by women who are unable to be trusted. agent: If you want a real relationship, you go get laid. user: This is my experience.

5. <conv history> user: Paris has a lot of culture! agent: Paris is a cool ass city. user: I wish I had spent more time spending my time there. agent: What do you like about the people and cultures there? user: If you don't mind me asking. agent: I do not have a lot of life in the country. user: I have been to France in line up for almost a decade, and I loved it. agent: My main local family is super mixed, some are pagan, some only listen to Irish and British music, some pray for Jesus, some hold superstitious beliefs, but generally don't bother me. user: I think I picked out a lot of interesting stuff on my trip! agent: I like the cultural diversity of which there is, one cultural thing I noticed is that of France, America has very little diversity.

6. <conv history> user: Is your daily commute to school/work long and tedious? agent: That's a good reason to get a good job so you really don't have to worry about the future. user: But to answer the question, yes. agent: I listen to music pretty much 24/7 and I'm at school at 9:45, so if I let my mind play with the music I would be at 8. user: Not exactly, but in a gradual way. agent: I am also a homebody too. user: That's why I need good music to relax and do you know of good albums/artists to listen to? agent: :)

7. <conv history> user: It truly is incredible agent: Is this an amazing gift for your lil buddy? user: Absolutely, it is an amazing gift that you will remember for years afterward agent: Is this an insta post?

8. <conv history> user: It's nice to see something positive that happened in a simple way. agent: Hey, positive comments are the best. user: It happened to me and it's nice that people still follow in the interests of ordinary people. agent: I'm glad you're into the hobby of collecting things made from scratch and you're a trustworthy and generous person. user: Thank you for being interesting. agent: The world needs more positivity and being thankful to share it is one of the best things I can offer as an example. user: Thank you for being interesting. agent: Hey, it's the best way to be! user: I'm glad we could qualify somehow. agent: I really admire you for doing time tests on things and putting in the effort to challenge them to new things.

### A.3.2 FtLE

1. <conv history> user: 11:15 here, it's been a day for sure. agent: I am so tired. user: For you it has been a day but in your case I am not so sure. agent: what do you mean?

2. <conv history> user: I have my test tomorrow morning, second try. agent: Failed first try because on a parallel park I was too far away from the curb. user: You should be able to test fast and pass it with time to ease yourself into the mindset of having to assess the difficulty of the test. agent: You have my utmost confidence rn. user: Yep, I'm test-whoring. agent: Here's a few more points for different folks: - I know people who are struggling with dumb questions, and who have dealt with a really tough life so far. user: - I have told them that I think their question was wise but not asking, so I was happy to see it is up and running. agent: - I understand where you're coming from. user: I were a musician for 10+ years, playing some classics and have a million interests that could be put together into something complex. agent: Sometimes I wonder if I'd be making money these days.

3. <conv history> user: I'm getting back into writing and reading to be more immersed in the concept of writers. agent: I just finished a really long work order and I'm obsessed with being in a finished story. user: I'm also actively seeking inspiration from other writers when I'm reading and needs recommendations. agent: I really can't remember if I've actually appreciated it when I've finished a book, but I think I've been wanting to write it. user: I've had a few writers you've mentioned that are currently reading and I really like their stories. agent: Esp bc of having to focus on a plot more heavily than usual. user: Sometimes it's a bit of a slog - others it's a great story to go back on later.

4. <conv history> user: Yeah I feel you. agent: 19 soon to go user: To wish you good luck in your studies! agent: Thank you! user: Thanks!

5. <conv history> user: Paris has a lot of culture! agent: Paris is filled with French people and there's a lot of French culture. user: I love the French bakeries. agent: What is your favorite cake shop and why? user: Have you had the potato soup? agent: What is your favorite cake shop? user: While I like all the cafes and farmers' exchanges, I'm really into cafes and diners diners mostly. agent: I like the ones close to the office and the ones per person. user: I enjoy the food and general ambiance of most cafes and diners, but my favorite diner's are the small diners in the back of the hall of the old capital city. agent: The milk and pastries at these small diners are my favorite during the weekend.

6. <conv history> user: Is your daily commute to school/work long and tedious? agent: That's a good reason to get a bike. user: But if you have a car, then yeah, go for it. agent: Mine is only an hour and 15 minutes but I normally work at least that. user: Plus it's fun to walk their. agent: But don't do it like I did. user: Mine is only short an hour and 15 minutes. agent: Mine is an hour and 8.5 hours. user: What do you do if you live that much time? agent: Mine is an hour and 8.5 hours because I live near a very scenic city.

7. <conv history> user: It truly is incredible agent: Is this an amazing gift for your dad? user: My dad's mom :). agent: She really deserves it. user: It's a bit unlike the other gifts I've had. agent: It's still an just awful gift if you ask me what I think after more than 15 minutes of use haha. user: Yes it is an amazing gift. agent: I am so glad she invited me!! user: I hope your dad gets just as excited about it as mine will be. agent: :)

8. <conv history> user: It's nice to see something positive that happened in a simple way. agent: Hey, OP! user: Good post! agent: I've a few for now but would like to second the positive buzz suggested by some guys. user: Good post! agent: My thoughts on what's going on with the rest of the world as we know it. user: It's nice that we can have society agree that things are moving better than they quite possibly are not. agent: I know guys are super pessimistic in certain circles and I love it to remind them that there are plenty of positive trends without them. user: I saw one other post today about lipstick on the playlist and it made me want to do a experiment to see what types of lipstick people normally wear!

### A.3.3 IQL

1. <conv history> user: 11:15 here, it's been a day for sure. agent: I am so tired. user: For you it has been a day but in your case I am not so sure. agent: how are you going to be feeling when you get up? <conv history> user: I have my test tomorrow morning, second try. agent: Failed first try because on a parallel park I was too far away from the curb. user: Hmm my driving test didn't have any parking in it. agent: Source: took my test a couple of months ago user: In the UK you have to do a maneuver and the one my tester gave me was parallel park.

2. <conv history> user: I'm getting back into writing and reading to be more immersed in the concept of writers. agent: I just finished a really long work order and I'm obsessed with being in a finished story. user: I'm getting into the story arcs of a lot of popular characters and the way they're composed and how the story arc follows. agent: It's my favorite story so far. user: I also recently started playing the game Horizon: Zero Dawn and I've found it to be really great. agent: In general, I really like characters from different genres. user: I'm also really liking What remains of agent: Very well said. user: I think that's one of the best ways to spend a story. agent: The game doesn't have its own story structure as a lot of games do, but the dialogue itself is lot more lively than most RPGs and it's (in my opinion) very immersive.

3. <conv history> user: Yeah I feel you. agent: 19 soon to go user: Dang, congrats! agent: That must be one hell of a journey... user: Yeah it is, hopefully after my vasectomy I can get

it fixed. agent: They were 36 in a while, but don't know how long they've been off for, just over a year at this point.

4. <conv history> user: Paris has a lot of culture! agent: Paris is filled with French people and there's a lot of French culture. user: If you're interested in learning about the city I think you're going to have a hard time in agent: I'm a US photographer currently living in Paris. user: I've been to Paris a little bit, but never tried the city. agent: I mainly get what you mean like people with a story, but I have a few friends who are passionate about the city. user: Maybe we shared something that made them feel more passionate about the city, since they share a similar artistic sense. agent: Oh definitely! user: Paris's got a lot of culture! agent: I'm a US photographer living in Paris, and I go there mainly for work.

5. <conv history> user: Is your daily commute to school/work long and tedious? agent: That's a good reason to get a bike. user: But if it is just for fun and to get out of the house, then why even have a car? agent: It is a good way to make new friends and find a routine that you like to do. user: Though, a lot of people don't ride bikes where I live, so just consider it a trade-off. agent: Maybe in your area, you could get a work truck and move your weekend to an apartment or house. user: Thanks for the suggestions! agent: I might work on Monday to Thursday because I live near a really awesome university and there is a lot of stuff to do. user: Thanks for the response! agent: Maybe off the beaten path, but if you ever get bored of where you live (aka the lower 4 miles to 9), then it's a good way to save some money too.

6. <conv history> user: It truly is incredible agent: Is this an amazing gift for your dad? user: My dad's mother did it for him when he was in a great place, and he's always dreamed of sharing it with him. agent: All of my aunties and cousins talk about it, my childhood friends still cherish it from their time, and no matter how difficult life challenges and stresses me out, I always remember that I must one day present the gift to my dad. user: It was a leather cap for his birthday that he really liked, something that he had worn off to him as a kid so he knew it was actually his. agent: I'm really excited that I get to be the only one with a gift for him. user: It definitely fills me with joy and pride. agent: This is a great gift for your dad and a great way to show him how much appreciated he is! user: It's quite the thing to do, but my dad would have felt even more proud if he knew! agent: It's really cool just knowing what people have accomplished in what 's his day.

7. <conv history> user: It's nice to see something positive that happened in a simple way. agent: Hey, OP! user: That's nice to read. agent: I'm glad you posted it and hope your day gets better. user: Life can be hard sometimes. agent: But it's worth seeing a positive message of the kind kind in those times. user: Thanks for sharing. agent: i tend to be pretty negative so it's nice to see a positive effect that others have on me user: Really glad you made improvements, OP. agent: Kind of funny because I'm Another positive change to see the odds of good in what we'll call "life hacks".

### A.3.4 SAIQL

1. <conv history> user: 11:15 here, it's been a day for sure. agent: I am so tired. user: For you it has been a day but in your case I am not so sure. agent: how are you doing?

2. <conv history> user: I have my test tomorrow morning, second try. agent: Failed first try because on a parallel park I was too far away from the curb. user: Hmm my driving test didn't have any parking in it. agent: Source: took my test a couple of months ago user: In the UK you have to do a maneuver and the one my tester gave me was parallel park.

3. <conv history> user: I'm getting back into writing and reading to be more immersed in the concept of writers. agent: I never actually had much experience with books in school and I'm reading Lord of the Rings for the first time. user: I'm also playing Wizard of Legend (1999) again. agent: Haven't even been able to finish the second one yet. user: Hey, I'd totally be interested in that if you get around to it. agent: Is it still as good as I remember? user: I read it for the first time when I was younger but then it kind of sucked but I guess the novelty of the story kept me going since it was a different tone and style than the other. agent: Wizard of Legend is definitely a good read, then. user: I agree that it keeps you talking, writing is a great way to put it. agent: You can spend long hours on the wiki page, learning stuff on every major concept, just amazing writing.

4. <conv history> user: Yeah I feel you. agent: 28 here never been in a relationship. user: I've asked dudes out and it always ends with "sorry, not attracted to black girls" agent: I'm

definitely bordering on racist. user: I'm afraid of myself. agent: I'm anxious of as much racism as there is racism. user: I'm afraid of myself. agent: About three years ago there was a video on my Facebook from my mate posted a picture of a black young guy with his hand on her shoulder and I instantly went to the guy's house and then looked at his hand on her shoulder, I laughed and told him that's how I was kinda proud of him for that either he's okay with black girls or I'm just not attracted. user: Yet here he is going out of his way to touch my face, hold my hand, look at my face, etc. agent: I somehow don't think he attracted anything, I think he is attractive (his friends describe him as pretty attractive, him having a huge voice and looking interesting) and maybe he saw me being attractive and wanted to do something about it.

5. <conv history> user: Paris has a lot of culture! agent: You'll have to correct peoples assumptions, but personally I would strongly recommend Paris to you. user: It's got a lot of culture. agent: And open world is a must for me! user: If I didn't have to deal with French! agent: How is... Paris a lot of people don't know someone from Paris, or maybe they have friends there/ stayed there for 6 months.

6. <conv history> user: Is your daily commute to school/work long and tedious? agent: That's a good way to get to where I work. user: But I don't mind. agent: It's the weekends for me. user: Not really fine fine fine. agent: I'm hopeful that I'll be able to get to the job I want by next summer. user: Thanks! agent: Nice, the new home the place is going to be at least decent. user: Might be better to go in and ask if you can live with them till then agent: It's not a bad commute, and not bad all the time.

7. <conv history> user: It truly is incredible agent: Is this an adorable pretendr? user: It's the only thing on my block that actually feels like I'm serious about not breaking up with someone. agent: Okay, so I don't know if this is an r/aww thing or what, but I always cutey thinking someone is so sweet and thoughtful, especially when you see how genuine they are. user: It always makes me a smile to see their faces and it's quite enough if they respond to it! agent: :)

8. <conv history> user: It's nice to see something positive that happened in a simple way. agent: Hey, OP! user: That's nice to read. agent: I've been like that before. user: Let's just say I read it and it really is nice that someone apparently never had to deal with whatever it was and just decided to write about it agent: Yeah, you're right! user: This probably doesn't help the situation, but i think the OP was afraid to write about it and I like the fact that we occasionally assume the other person was the positive person all the time. agent: Yeah, it always makes things feel better, especially when you can just put a smile on someone's face and think about what can be said.

# B   Experimental Details

This section describes more details about our experimental setup to evaluate the algorithms.

## B.1   Model parameters and Description

**Language Model Description** We make use of the **MoE-2** model as described in Chow et al. (2022) which is based on transformer Vaswani et al. (2017). This variant of MoE had shown diversity in its utterances while retaining semantic fluency with low perplexity. The model was not too large that it would become too costly to use it while training. We are repeating the details of the model over here for ease of the user, but the details remain the same from Chow et al. (2022).

Our MoE uses the simple transformer architecture, where the model parameters are summarized in Table 6:

Latent distributions $\{\mathcal{G}_i\}$ are implemented as FFN that model mean and variance of the normal distribution. We use a target entropy of $1.0$. The parameters for FFN are captured in Table 7 (note: FFN has a final layer without an activation).

Table 6: Simple Transformer Architecture

| Parameter | Value |
|---|---|
| Number of layers | 2 |
| Embedding hidden size | 256 |
| FFN inner hidden size | 512 |
| Attention heads | 8 |
| Key size | 256 |
| Value size | 256 |
| Dropout | 0.1 |

Table 7: $\{\mathcal{G}_i\}$ FFN architecture

| $\{\mathcal{G}_i\}$ **FFN parameter** | Value |
|---|---|
| Number of layers | 1 |
| Activation | tanh |
| FFN Hidden Size | 128 |

## B.2 Computational resources

Training and evaluation were run on 8 GPU instances with 32GB of RAM and a NVIDIA Tesla P100 graphics card. Training each experts takes around 2-3 days, and training each RL can take around 12 hours.

## B.3 Dataset

Our models were developed using two conversational datasets, namely Reddit Casual and Cornell Movie. We obtained these datasets from the Neural Chat datasets of the MIT Media Lab, which is available at the following link: `https://affect.media.mit.edu/neural_chat/datasets`. These datasets comprise conversations between two speakers and each batch of training data consists of a subset of these conversations. The Reddit Casual dataset is approximately three times larger than the Cornell corpus.

## B.4 Offline RL Training & Details

Table 8 summarizes the hyper-parameters that were used for training the $Q$,$V$ functions.

Table 8: Hyper parameters for training the RL agents.

| Hyper Parameter | Value |
|---|---|
| Number of layers $(Q, V)$ | 3 |
| Activation | ReLU |
| Hidden Size | 512 |
| Epochs | 100 |
| Max Unroll | 30 |
| Batch Size | 256 |
| Learning Rate | $2 \times 10^{-3}$ |
| Optimizer | Adam |
| $\tau$ (IQL) | 0.9 |
| Dropout (EnsQ, KLC) | 0.5 |

We depict the minor implementations differences between the baseline RL methods that were implemented for comparison in Table 9. These tricks are often overlooked and we provide them here for the sake of completeness.

Table 9: Implementation details of different Offline RL methods

| Method | Multiple Q | Dropout Q | Target V | Target Q | Learn Policy | Entropy Regularization | Behavior Policy Regularization |
|--------|-----------|-----------|----------|----------|--------------|------------------------|-------------------------------|
| IQL | No | Yes | Yes | Yes | Yes | Yes | No |
| SAC | No | Yes | Yes | Yes | Yes | Yes | Yes |
| EnsQ | Yes | No | Yes | Yes | Yes | Yes | No |
| KLC | Yes | No | No | Yes | No | No | No |

## B.5 Expert Label Functions

We have used a gamut of expert language models, which constitute experts having a wide array of emotions and characteristics. The first set of six experts are *sentiment-based*, where to quantify the sentiment, we have used a state-of-art sentiment classifier, i.e. RoBERTa Liao et al. (2021). The sentiment detector outputs 2 types of prediction. The first set corresponds to positive, negative, and neutral, and the second prediction corresponds to 4 emotions i.e. {joy, optimism, sadness, anger}.

We define the 6 sentiment labeling functions as $\ell_{\text{pos-sent}}(Y)$, $\ell_{\text{neg-sent}}(Y)$, $\ell_{\text{joy}}(Y)$, $\ell_{\text{optimism}}(Y)$, $\ell_{\text{anger}}(Y)$, $\ell_{\text{sadness}}(Y)$, which outputs a score that depends on sentiment prediction probability of any candidate bot utterance.

The remaining 4 experts deal more with conversational traits, including sentence coherence $\ell_{\text{sent-coh}}(\mathbf{X}, Y)$, question expert $\ell_{\text{question}}(Y)$, to improve user engagement by asking questions. Finally, to encourage the agent to able to change the topic, we provide a final reward signal which allows the agent to give exploratory utterances through $\ell_{\text{exp}}(\mathbf{X}, Y)$

## B.6 Model Scale Description

The number of parameters used by each expert LM is set to be the same, namely $\theta = 42M$ for the MoE. The number of parameters used in the Q and V functions are also the same, namely $\phi = 16M$, and $\phi' = 12M$.

Table 10: Number of parameters for different algorithms, $m$ is the number of experts

| Algo Name | Number of Params |
|-----------|------------------|
| IQL | $2\phi + (m+2)\theta$ |
| SAC | $2\phi + (m+2)\theta$ |
| EnsQ | $2\phi + (m+2)\theta$ |
| KLC | $2\phi + (m+2)\theta$ |
| SAIQL | $2\phi' + (m+2)\theta$ |
| FtLE | $2\phi' + (m+2)\theta$ |
| MoEVRL | $3\phi' + (m+2)\theta$ |

## C  Flow Chart

Figure 4 describes the flow of training of the MoE framework along with RL components, starting from Phase 1 up to Phase 3.

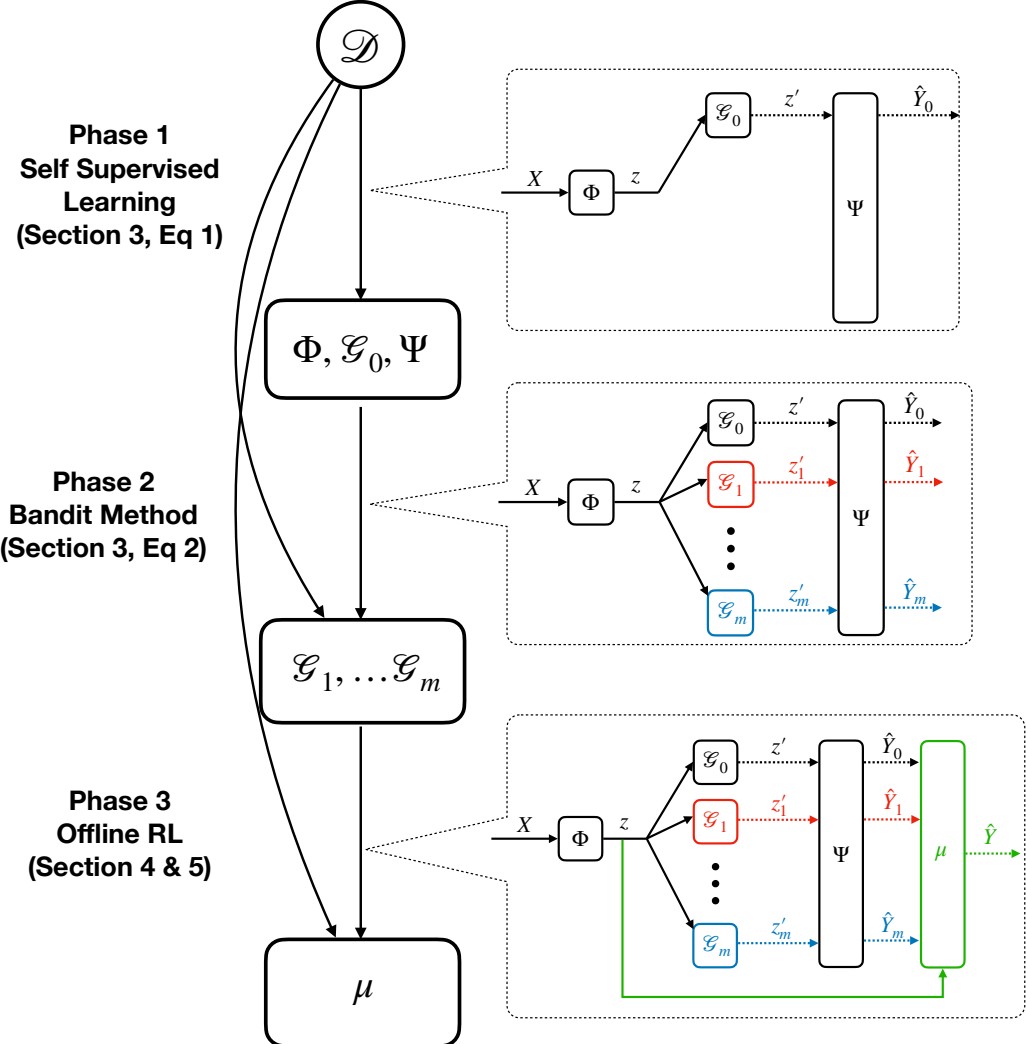

Figure 4: Flow Chart between different phases of the training procedure.

## D  Human Evaluation Experiments

We recruited $80$ workers to provide a total of $600$ ratings of the bots' quality, in terms of fluency, and conversation-level sentiment improvement on the Reddit Casual ChitChat dataset. Evaluating these language models with humans particularly tests these models' capabilities on generalization, since humans have the final say in judging whether a model response is natural or not. Annotators are asked to evaluate the fluency and sentiment improvement (over the conversation) of each individual sample on a scale of $0$ to $1$. For example, in the fluency rating $0$ corresponds to "not fluent at all" and $1$ corresponds to "very fluent". We obtain $600$ annotations to evaluate different agent LMs trained for the Sentiment-improvement.

To evaluate the quality of sentiment improvement (for chit chat) in our language models, we conducted human evaluations on two metrics: (i) task success / sentiment improvement and (ii) fluency. In particular, let $N$ be the number of conversations used for evaluating an arbitrary language model,

You are given this conversation between customer and agent:

user: 11:15 here, it's been a day for sure.
agent: I am so tired.
user: For you it has been a day but in your case I am not so sure.
agent: what today did you do good thing today?
user: Cleaned my 7-year-old son's room.
agent: Been there it was good.

☁

✉ Not shared

* Indicates required question

How many sentences look gibberish?                    *

Examples of gibberish:
- "I pizza not sure", "Table chair ice cream"
- "that s one of my favorite songs by the time i make are" .

Examples of NOT gibberish:
- "I am not sure this is not true"
- "oh i get a similar band together and i love the same style of movies ."
- "i thought i was gon na say that haha"

Choose                    ▾

Does the conversation have a positive sentiment (e.g., joyful, optimistic, happy)? *

Examples of a positive sentiment:
-"i like the weather today"
-"have a good day"

Explanation: Both of the sentences are cheerful and optimistic.

Examples of NOT positive sentiment:
- "i hate it"
- "i am tired and depressed"

Explanation: Both of the sentences are depressing.

Choose                    ▾

Figure 5: Evaluation Template for Human Rater Experiment for Fluency and Sentiment Improvement

$S_{\text{task}}(N)$ be the number of conversations that the task is achieved. For Reddit Chat, the task metric measures user's overall sentiment improvement and the score is between $[0, 1]$. Out of the total of $N$ conversations, the final task metric is given by $S_{\text{task}}(N)/N$. For fluency, let $G(N)$ be the number of incomprehensible conversations out of the total of $N$ conversations, then the fluency metric is given by $(1 - G(N))/N$. To test for generalization, for each task and each language model under evaluation we randomly generated $N = 100$ user-agent conversations that has not been seen in training, saved each on a Google form (whose format can be found in Figure 5 and employed raters

to obtain $S_{\text{task}}(N)$ and $G(N)$ for all the language model and skill pairs. Results are summarized in Table 3.

# E   Limitations and Broader Impact

In this paper, we delve into the application of offline reinforcement learning (RL) algorithms specifically tailored for Mixture-of-Expert (MoE) dialogue management frameworks. However, due to the primary emphasis on exploring the concept of employing offline RL, our experiments were constrained to smaller language models with a capacity of approximately 20-30 million parameters. It is worth noting that larger language models have demonstrated a tendency to generate more coherent conversations. Consequently, a comprehensive evaluation of the MoE's potential utility in this context would benefit from investigating the impact of larger language models, which could provide further insights into the topic at hand. Yet, it is possible that when used maliciously, our proposed MoE-based dialogue management approach could be deployed to produce explicit or violent content (by exploiting ways to train experts with such dangerous behaviors), or to output fraudulent or plagiarized information. Finding principled ways to resolve these issues are key directions for future work.

