# Offline Reinforcement Learning for Mixture-of-Expert Dialogue Management

## Abstract

Reinforcement learning (RL) has shown great promise for developing dialogue management (DM) agents that are non-myopic, conduct rich conversations, and maximize overall user satisfaction. Despite recent developments in RL and language models (LMs), using RL to power conversational chatbots remains challenging, in part because RL requires online exploration to learn effectively, whereas collecting novel human-bot interactions can be expensive and unsafe. This issue is exacerbated by the combinatorial action spaces facing these algorithms, as most LM agents generate responses at the word level. We develop a variety of RL algorithms, specialized to dialogue planning, that leverage recent Mixture-of-Expert Language Models (MoE-LMs)—models that capture diverse semantics, generate utterances reflecting different intents, and are amenable for multi-turn DM. By exploiting MoE-LM structure, our methods significantly reduce the size of the action space and improve the efficacy of RL-based DM. We evaluate our methods in open-domain dialogue to demonstrate their effectiveness w.r.t. the diversity of intent in generated utterances and overall DM performance.

## 1 Introduction

Natural Language Processing (NLP) has made significant strides in recent years, notably in the field of language generation. Thanks to advances in language modeling, particularly with the use of transformer Vaswani et al. (2017), NLP models can now generate human-like text that is often difficult to distinguish from text written by a person. However, despite these advancements, these models still fall short when it comes to having rich conversations. Current NLP models lack effective dialogue management, as these models are good at generating individual sentences, but struggle with maintaining coherent and engaging conversations. Whereas, most compelling conversations generally span numerous topics, are rather open-ended, and often have an underlying goal (e.g., customer success, task completion, recommendation). This requires dialogue agents to understand the context of the conversation and respond appropriately while maintaining the ability to achieve goals.

*Reinforcement learning (RL)* is a natural approach for optimizing a dialogue management agent's policy. Earlier work on RL-based dialogue systems relies on specific, hand-crafted semantic states (Levin and Pieraccini, 1997; Singh et al., 2002; Walker, 2000) or partially observable belief states (Williams and Young, 2007; Young et al., 2010), in which case the agent encodes conversations and chooses the best structured dialogue action at each turn. Applications include relational reasoning (Shah et al., 2018), task completion (Shi and Yu, 2018), and query fulfillment (Serban et al., 2017), whose action spaces are structured enough to be represented by hand-crafted features. To handle more complex dialogues, recent approaches use language models to extract semantic representations from conversation histories, treat these representations as dialogue states, and apply RL to learn a word-level generative DM agent (Jaques et al., 2019; Li et al., 2016, 2017; Shin et al., 2020).

However, unlike supervised learning approaches, where one can train imitation agents with offline conversation data, RL DM algorithms require online exploration to learn effectively. Unfortunately,

constant interactions with real users is often expensive and time-consuming. While one can potentially address the DM problem using *offline* RL, issues such as model exploitation leading to distribution shift on the state and action space, when training on static datasets are of paramount concern (Levine et al., 2020). Moreover, the myriad variation of language makes incorporating all possible conversation histories and bot utterances into the state and action spaces of an RL formulation of the DM problem impractical due to the combinatorics at play. As a result, naively applying RL to DM may result in poorly-performing agents that generate incomprehensible utterances (Zhao et al., 2019).

We tackle the issues above, related to the use of offline RL in DM systems, by leveraging recent advances in Mixture-of-Expert Language Models (MoE-LMs) (Chow et al., 2022). Specifically, we develop a suite of offline RL algorithms specialized in dialogue planning that exploit the structure of MoE-LMs. Our methods consist of three main components: **1)** a primitive LM which, using a probabilistic encoder and decoder, is capable of generating diverse semantic intents **1)** a primitive LM that uses a probabilistic encoder-decoder pair to generate sentences with diverse semantics and intents ; **2)** a number of *specialized* expert LMs, each of which generates utterances corresponding to a specific intent; and **3)** a compositional dialogue manager (DM) that, at each turn, given the encoded conversation history and a set of candidate utterance suggested by the experts, selects one candidate utterance for the DM agent to execute as a response to the conversation until now.

Our contributions to offline RL adapted for MoE-based DM agents are four-fold. First, we exploit the hierarchical structure of MoE-LMs, allowing our offline RL methods to work with a significantly smaller, finite action space, hence making the RL problem more tractable. Second, by leveraging pre-trained MoE-LMs—which generate sensible utterances—and offline RL *prior regularization*—which matches our DM's behaviors with that of the primitive LM—our RL algorithms focus on higher-level dialogue planning, and are more data-efficient than standard RL methods by allowing

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

 and intent. For dialogue management, the compositional DM $\mu$ takes as input the

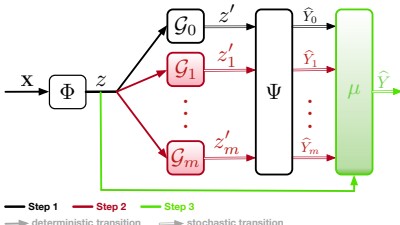

Figure 1: MoE-LM Architecture.

encoded conversation history $z = \Phi(\mathbf{X})$ and candidate action utterances generated by the experts $\{\widehat{Y}_i\}_{i=0}^m$, and selects one of them to execute, i.e., $Y \sim \mu(\cdot \mid z, \{\widehat{Y}_i\}_{i=0}^m)$. Given the state $s = \mathbf{X}$ and action $a = Y$, the MoE-LM policy that optimizes the DM MDP can be expressed as

$$\pi_{\text{MoE}}(a|s) = \int_{\{\hat{a}_i, z_i'\}_{i=0}^m} \mu(a|\Phi(s), \{\hat{a}_i\}_{i=0}^m) \prod_{i=0}^m d\Psi(\hat{a}_i|z_i') d\mathcal{G}_i(z_i'|\Phi(x)). \tag{1}$$

# 3 Warmstarting the MoE-LM

The MoE-LM approach reformulates the RL dialogue management problem with much smaller state and action spaces and focuses on optimizing the specific goal of the conversation task (as candidate utterances are separately optimized to follow particular bot-based characteristics/intents). Recall that the DM is a policy conditioned on both the latent state and the actions suggested by the experts. Before introducing the different RL methods for DM (Section 4 and 5), in the following we outline (i) the learning of diverse semantics (primitive LM) for conversation histories, which allows the agent to generate a wide variety of utterances, and (ii) the construction of specialized LMs (experts), which generate utterances of different intents.

Following from the primitive discovery procedure in Chow et al. (2022), the primitive LM, $\text{LM}_0$, is learned by solving a KL-constrained optimization problem that aims at capturing diverse semantics:

$$\min_{(\Phi, \mathcal{G}_0, \Psi), \rho} \widehat{\mathbb{E}}_{z' \sim \rho(\cdot|z, Y), z = \Phi(\mathbf{X})} [-\log \Psi(Y|z')] \text{ s.t. } \widehat{\mathbb{E}}_{z = \Phi(\mathbf{X})} [\text{KL}(\rho(z'|z, Y) || \mathcal{G}_0(z'|z))] \leq \epsilon_{\text{KL}}, \tag{2}$$

where $\widehat{\mathbb{E}}$ is the empirical expectation over $(\mathbf{X}, Y)$ in the dataset $\mathcal{D}$, $\rho$ is a distribution over the latent space conditioned on the encoded conversation history $z$ and the target utterance $Y$, and $\epsilon_{\text{KL}}$ is a positive real-valued threshold. Using (2), we learn $\text{LM}_0 = (\Phi, \mathcal{G}_0, \Psi)$ by maximizing the log-likelihood of sentence $Y$ for a context and latent generation, while enforcing consistency between the latent variable $z'$ predicted by $\mathcal{G}_0(\cdot|z)$ and $\rho(\cdot|z, Y)$ via the KL constraint. The distribution $\rho(\cdot|z, Y)$ is a Gaussian $\mathcal{N}\big(\mu_\rho(z, \Phi_\rho(Y)), \sigma_\rho^2(z, \Phi_\rho(Y))\mathbf{I}_{d \times d}\big)$ in which $\Phi_\rho$ is a pre-trained encoder for the target utterance $Y$, and where the mean $\mu_\rho(\cdot, \cdot)$ and the variance $\sigma_\rho^2(\cdot, \cdot)$ are trainable models. In practice, we implement the KL constraint in (2) as a penalty weighted by a chosen coefficient.

To complete the MoE framework, one needs to train a set of experts $\text{LM}_i$, $\forall i \in \{1, \dots, m\}$, with each generating candidate utterances of different intents. By viewing each expert as a distribution of particular behaviors in conversation data $\mathcal{D}$, we leverage the results in Chow et al. (2022) and adopt a universal encoder-decoder $(\Phi, \Psi)$ among all the experts. Therefore, each expert $i$ is parameterized by an arbitrary latent distribution that samples certain regions of the latent space $\mathcal{Z}$. Let $\ell_i(\mathbf{X}, Y) \in \mathbb{R}$ be a real-valued label that *characterizes* the intent of expert $i \in \{1, \dots, m\}$. We can think of $\ell_i(\mathbf{X}, Y)$ as score assigned to $Y$ resembling how strongly $Y$ exhibits the trait expert $i$ is meant to represent. We train the latent distribution $\mathcal{G}_i(z)$ of expert $i$ by solving the problem

$$\min_{\mathcal{G}_i} \ \widehat{\mathbb{E}}_{z' \sim \mathcal{G}_i(\cdot|z), z = \Phi(\mathbf{X}), Y \sim \Psi(\cdot|z')}[-\ell_i(\mathbf{X}, Y)]. \tag{3}$$

Each expert is learned via *reward-maximization*, where $\ell_i$ is treated like a reward signal w.r.t. expert $i$, wherein the expert tries to maximize that intent-aligned reward. Note that there is a correspondence of the above approach with contextual bandits (Chu et al., 2011), for which both the context and action spaces are latent space $\mathcal{Z}$, and the bandit policy is the latent distribution $\mathcal{G}_i$. The choice of greedy reward maximization is to encourage a particular behavior in the expert's immediate utterance rather than trying to control future utterances. Long-term dialogue planning is handled by the compositional dialogue manager. For example, with Gaussian experts $\mathcal{G}_i$, $i \in \{1, \dots, m\}$, we can use the standard REINFORCE (Sutton et al., 1999a) algorithm where the model parameters $(\mu_i, \sigma_i)$ are updated in the following direction, where $\alpha > 0$ is the learning rate – $\alpha \cdot \mathbb{E}_{z' \sim \mathcal{G}_i(\cdot|z), Y \sim \Psi(\cdot|z')}[\ell_i(\mathbf{X}, Y) \cdot \nabla_{\{\mu_i, \sigma_i\}} \log \mathbb{P}_{\mathcal{G}_i}(z'|z)]$. To reduce the variance of these estimates, we can also adopt the baseline reduction technique in (Greensmith et al., 2004).

## 4  RL for Mixture-of-Expert DM

Offline RL, in which the policy must be learned from the collected conversations $\mathcal{D}$ (without further online interactions), potentially allows RL DM methods to leverage the abundance of offline conversational data for policy learning. Denote by $(\mathbf{X}, Y, X_+) \sim \mathcal{D}$ a tuple sampled from the offline conversation data $\mathcal{D}$, where $X_+$ is the follow-up user response, and where $s := \mathbf{X}$, $a := Y$, $r(X_+)$, $s_+ := (\mathbf{X}, Y, X_+)$ are

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

demonstrate the diversity of intents in different offline RL algorithms in the model-based evaluation of the Cornell dataset. Given approximately 200 conversation turns, we measure the frequency of the expert agents when their utterances are selected and preset such frequency metric for the worst performing Offline RL method (**EnsQ**), a good performing method (**IQL**), and an MoE-specific RL algorithm (such as **MoE-VRL**). To visualize our findings, we plot a histogram of the frequencies on different experts being selected and calculate the KL divergence distance of this histogram and a uniform distribution over the experts. While we acknowledge that a uniform distribution may not be the optimal distribution of utterances, it provides a measure of how well the agents make use of different experts, along with their actual performance.

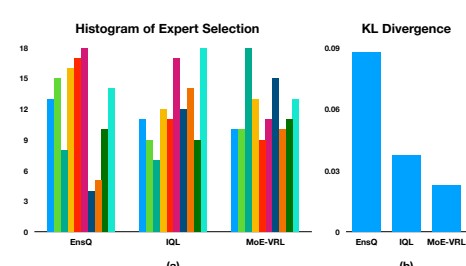

Figure 2: Experiment on the Cornell dataset with Model-based evaluation(a) Histogram of frequency of expert selection. (b) KL divergence against a uniform distribution

The results of Experiment 3 are shown in Figure 2, where we plot the frequency histogram of different expert agent utterances. We observe that the worst performing agent, **EnsQ**, has a highly skewed distribution of expert selections, with a few experts being heavily favored over others. This suggests that **EnsQ** is less diverse and does not effectively utilize the full range of expert knowledge available. On the other hand, both **IQL** and **MoE-VRL** exhibit a more balanced distribution of expert selection, with utterances chosen from multiple experts throughout the conversation; i.e., their frequency distributions are closer to a uniform distribution, with much lower KL divergence distance.

However, there is a clear performance gap between the two methods, with **MoE-VRL** significantly outperforming **IQL**.

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

# C   Use Case Figure

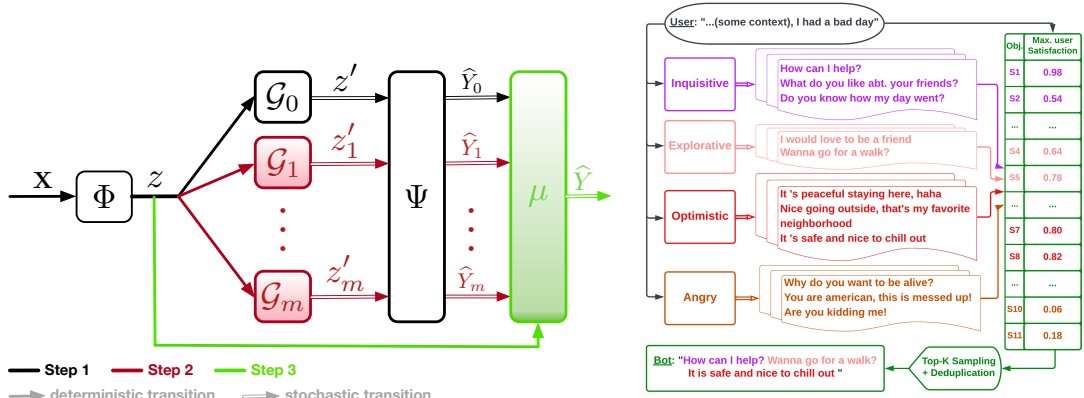

Figure 4: (Left) MoE-LM Architecture. (Right) Sample utterance workflow generated by an MoE-LM trained with Reddit data. Step 1: $\Phi$ encodes conversation history. Step 2: $\Psi \circ \mathcal{G}_i$, $\forall i$, generate candidate bot utterances. Step 3: $\mu$ selects the bot response by $Q$-score ranking & post-processing.

# D   Flow Chart

Figure 5 describes the flow of training of the MoE framework along with RL components, starting from Phase 1 up to Phase 3.

# E   Human Evaluation Experiments

We recruited $80$ workers to provide a total of $600$ ratings of the bots' quality, in terms of fluency, and conversation-level sentiment improvement on the Reddit Casual ChitChat dataset. Evaluating these language models with humans particularly tests these models' capabilities on generalization, since humans have the final say in judging whether a model response is natural or not. Annotators are asked to evaluate the fluency and sentiment improvement (over the conversation) of each individual sample on a scale of $0$ to $1$. For example, in the fluency rating $0$ corresponds to "not fluent at all" and $1$ corresponds to "very fluent". We obtain $600$ annotations to evaluate different agent LMs trained for the Sentiment-improvement.

To evaluate the quality of sentiment improvement (for chit chat) in our language models, we conducted human evaluations on two metrics: (i) task success / sentiment improvement and (ii) fluency. In particular, let $N$ be the number of conversations used for evaluating an arbitrary language model, $S_{\text{task}}(N)$ be the number of conversations that the task is achieved. For Reddit Chat, the task metric measures user's overall sentiment improvement and the score is between $[0, 1]$. Out of the total of $N$ conversations, the final task metric is given by $S_{\text{task}}(N)/N$. For fluency, let $G(N)$ be the number of incomprehensible conversations out of the total of $N$ conversations, then the fluency metric is given by $(1 - G(N))/N$. To test for generalization, for each task and each language model under evaluation we randomly generated $N = 100$ user-agent conversations that has not been seen in training, saved each on a Google form (whose format can be found in Figure 6 and employed raters to obtain $S_{\text{task}}(N)$ and $G(N)$ for all the language model and skill pairs. Results are summarized in Table 10.

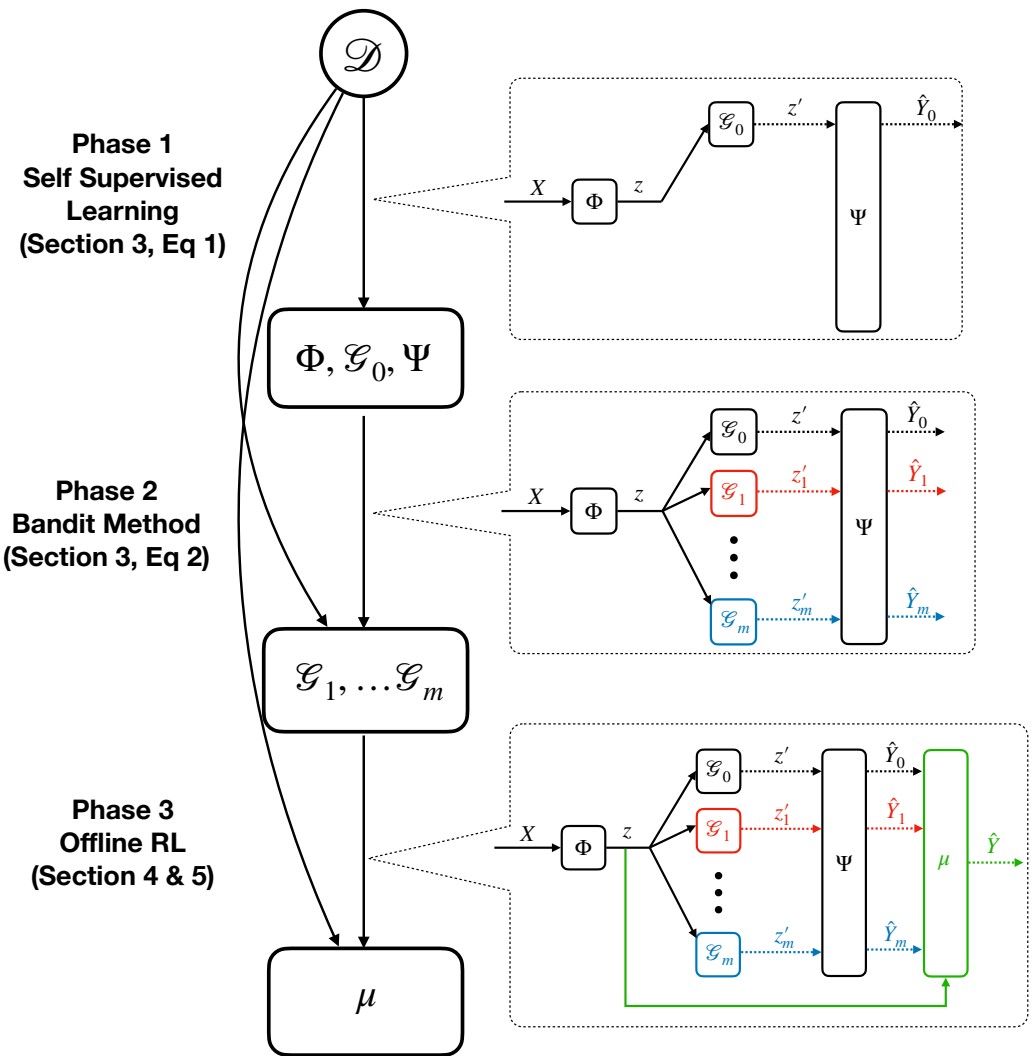

Figure 5: Flow Chart between different phases of the training procedure.

| Method | Avg. Fluency | Sentiment |
|--------|--------------|-----------|
| BC | $0.67 \pm 0.26$ | $0.24 \pm 0.50$ |
| KLC | $0.62 \pm 0.27$ | $0.66 \pm 0.47$ |
| IQL | $0.84 \pm 0.24$ | $0.72 \pm 0.46$ |
| SAIQL | $0.81 \pm 0.19$ | $0.57 \pm 0.50$ |
| FtLE | $0.88 \pm 0.24$ | $0.76 \pm 0.48$ |
| MoE-VRL | $0.72 \pm 0.28$ | $0.70 \pm 0.45$ |

Table 10: RL (Phase 3) Raters Evaluation

## F   Limitations and Broader Impact

In this paper, we delve into the application of offline reinforcement learning (RL) algorithms specifically tailored for Mixture-of-Expert (MoE) dialogue management frameworks. However, due to the primary emphasis on exploring the concept of employing offline RL, our experiments were constrained to smaller language models with a capacity of approximately 20-30 million parameters. It is worth noting that larger language models have demonstrated a tendency to generate more coherent

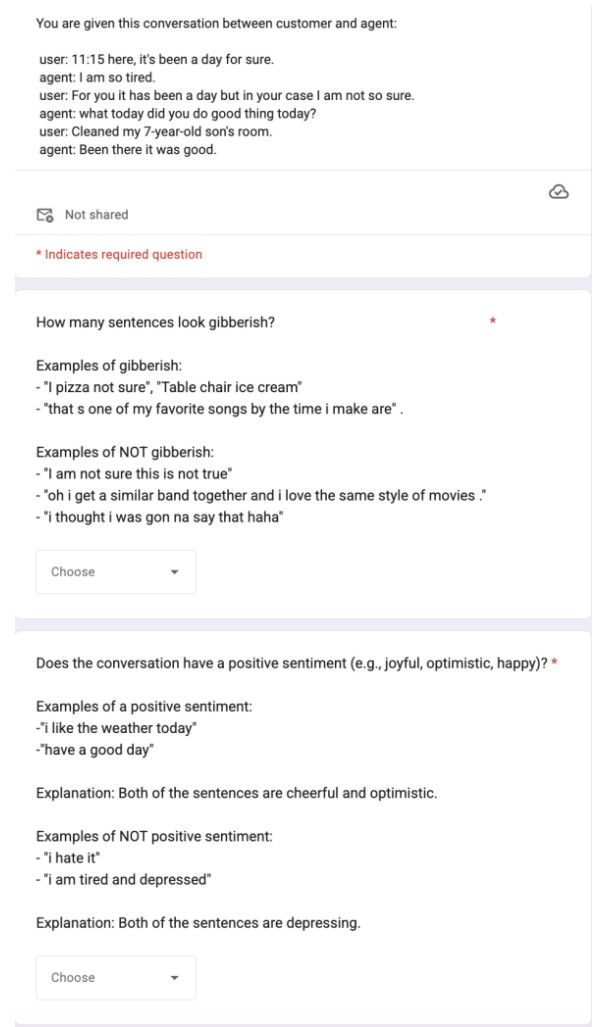

Figure 6: Evaluation Template for Human Rater Experiment for Fluency and Sentiment Improvement

conversations. Consequently, a comprehensive evaluation of the MoE's potential utility in this context would benefit from investigating the impact of larger language models, which could provide further insights into the topic at hand. Yet, it is possible that when used maliciously, our proposed MoE-based dialogue management approach could be deployed to produce explicit or violent content (by exploiting ways to train experts with such dangerous behaviors), or to output fraudulent or plagiarized information. Finding principled ways to resolve these issues are key directions for future work.