# OpenReview forum: "Offline Reinforcement Learning for Mixture-of-Expert Dialogue Management"
_NeurIPS.cc/2023/Conference — NeurIPS 2023 poster_

### Official Review · Reviewer_BhVQ · 2023-07-05

**Soundness:** 3 good
**Presentation:** 3 good
**Contribution:** 2 fair
**Rating:** 6
**Confidence:** 2

**Summary:**

This paper tackles issues with using RL for dialogue management related to covariate shift when using offline RL and the requirement for many online human-bot interactions when using online RL, and both suffering from a large action space. Current NLP models lack the ability to plan dialogue interactions beyond the next interaction, which is a crucial aspect of successful conversation. To address these points, the authors propose a mixture-of-expert language model (MoE-LM) and design several offline RL algorithms with different benefits. The structure of the MoE-LM is a hierarchical one, where there are a number of experts each optimised for a different intent (things like empathy, rage, etc.) and a dialogue management model that chooses one of the expert utterances conditioned on the conversation history. This latter aspect should help with non-myopic objectives that are naturally part of conversations. The authors compare their MoE-LMs trained with several different algorithms to SotA offline RL algorithms as well as behavior cloning and a bandits method that greedily optimises the next conversation turn (just reward maximisation). On two datasets, they show their method outperforms the offline RL baselines in terms of return in a simulated conversation (DialoGPT). They also show that the best performing RL algorithm for MoE-LMs which they propose has a more uniform selection of experts than the worst performing one.

**Strengths:**

Very clear and detailed explanation of the algorithms, well-motivated and important problem. Strong results of proposed method compared to baselines on two different dialogue tasks.

**Weaknesses:**

Some of the claims seem to not be substantiated by the results of the experiments:
- It seems likely that the MoE-LMs generate more diverse conversations, but strictly you can't assume that based on a higher return in terms of sentiment; can you supplement the results with metrics that evaluate diversity?
- It's not clear to me from experiment 2 that the MoE method is able to do long-term planning. Again, it seems likely given the performance increase over the offline methods and especially the Bandit, but simply based on the higher return in sentiment this can't be claimed. Can you isolate / evaluate the effect of long-term planning? How do we know from these results that the method is actually better at long-term planning and the performance increase is not due to other aspects that result in a better sentiment?
- I might have missed it but there does not seem to be an explicit treatment of the sample-efficiency of your method over others; can you quantify this?

It's pretty hard to interpret these results based solely on sentiment calculated over conversations with a simulated user. A baseline that would help interpret this to some extent is a simple prompted LLM to hold these conversations; what kind of sentiment would it achieve? Additionally, can you do a small human eval comparing your method to bandits and the best-performing offline RL method? Perhaps this can be combined with the questions above about diversity and long-term goal achieving / planning, asking humans to rate the conversations along these axes.

**Questions:**

Answers to the questions written in Weaknesses would address my main concerns with this paper.
Some small other questions/suggestions that are not related to my score:
- I don't really follow contribution 2; how does leveraging pre-trained LMs and prior regularization result in high-level dialogue management?
- I would rewrite line 58-62, very hard to parse with usage of "-- --" twice
- Line 102 should be textual citation (citet not citep)
- Table 1 and 2 need more comprehensive captions. Without reading the main text a reader should understand what the numbers and error bars refer to. What are your methods, what are the takeaways from these numbers, etc.

**Limitations:**

There's no separate limitations section or discussion of the limitations of your proposed method; that would be a welcome addition.

---

> ### Author Rebuttal · Authors · 2023-08-09
>
> We would like to thank the reviewer for the useful feedback. We will address the individual comments in the following shortlist of responses.
>
> ### Diversity of MoE responses; Diversity metrics
> We concur with the reviewer that high sentiment improvement in a conversation doesn't necessarily equate to utterance diversity. Though our return metric doesn't directly promote diversity, it is facilitated by the diverse semantic representation space and various candidate utterance generators within the pretrained MoE LMs. Empirically, as seen in Figure 2 and 3, the MoE-specific offline RL methods, while scoring some of the best returns, also tend to select responses from more diverse experts. This implies a greater variety in utterances. Conversely, utterance diversity doesn't guarantee high conversation return. For instance, the agent trained with the KLC offline RL method produces diverse responses but ranks among the worst performers, as shown in Table 1.
>
> ### Can the MoE method perform long-term planning; How to justify the efficacy of long-term planning
> The reviewer astutely questioned the long-term planning capabilities of the MoE-based RL method. Initially, we didn't compare with bandit-based baselines, as the advantages of RL-based methods for long-term planning are well-established in dialogue management (see Snell et al., 2023, Jaques et al., 2020). We followed similar considerations to Jaques et al., 2020, using the same datasets.
>
> To address the question, we evaluated the MoE-VRL offline RL method against a bandit agent, using a discounting factor of 0.0 (gamma=0.0), on the Cornell dataset. The 5-step cumulative model-based return for the bandit agent was **1.53**, significantly lower than the MoE-VRL results at a discounting factor of 0.8 (**3.62**). This experiment highlights the effectiveness of RL-based MoE dialogue management for long-term planning. We will include a discussion and the results in the final paper to highlight the benefit of long-term planning.
>
> ### Sample efficiency
> The improved efficiency arises from both modeling and algorithmic choices. First,  we adopt the MoE LM framework (Chow et al., 2023) and greatly simplifying the action space of the RL dialogue management problem, as we no longer rely on RL to directly control the token-level autoregressive generation of the language model but rather to select the best utterance to output at the current conversation turn from the pool of candidate responses generated by MoE-based model. Second, we developed our offline dialogue RL planning algorithms for MoE-LMs under the IQL offline RL methodology, which has shown in Snell et al., 2023  to have improved performance. While these factors intuitively improve sample efficiency, we acknowledge the reviewer’s comment that explicit analysis have not been conducted in our paper, and therefore we will update the conclusion section in the final paper to clarify these points and soften our claims about sample efficiency improvement.
>
> ### Baseline comparisons with prompting LLMs
>
> Comparing the responses of our MoE-LM agents with the ones generated by prompting LLMs is an interesting direction for future research.
>
> We would like to bring to the reviewer's attention that (i) rather than developing SOTA chatbots, the motivation of the work is to research different offline RL methods that make MoE LMs effective for multi-turn dialogue management; and (ii) the MoE LMs in our experiments are much smaller (~42M parameters) than standard LLM-based chatbots that generate diverse responses with various persona (e.g., a full GPT2 model has 1.5B parameters). With such a difference in model sizes, one may not expect the current MoE LMs can match the behaviors of any commercialized chatbots. On the other hand, it is already quite impressive to see that MoE LMs do possess different language skills, persona and have the capabilities to “smartly” switch among different language skills to improve the conversation.
>
> ### Human evaluation on diversity and RL performance
> Appendix E provides a human evaluation of different offline RL methods w.r.t to fluency and sentiment improvement in the overall conversation. Performance comparison of RL-based agents and the myopic, bandit counterpart has only been done via their corresponding cumulative returns (see above comments). We will add the human evaluation of the bandit agent in the final paper.
>
> ### Contribution 2
> This contribution summarizes our arguments about the sample efficiency (see above comments) improvements of specialized offline RL methods for dialogue management with MoE-LMs, via leveraging the MoE structure to simplify the RL dialogue planning problem and utilizing specialized offline RL methods to better solve this problem. We will clarify the presentation of contribution 2 in the introduction by including the above explanations.
>
> ### Table 1 & 2
> We acknowledge the reviewer’s confusion caused by the condensed presentation of these tables and will include more detailed explanations in the final paper. These tables present the average return (discounted sum of per-turn rewards in the dialogue conversations) of the dialogue agent of interest, accumulated over a 5-turn conversation. The return is averaged over 100 conversations, and the standard error is also provided.  A higher value indicates the corresponding agent is able to perform better dialogue planning and results in better overall sentiment improvement.
>
> ### Formatting issues
> Thanks for suggesting several modifications to improve the readability of our paper, we will incorporate them in the final draft.
>
>
> ### References
> Snell, C., Kostrikov, I., Su, Y., Yang, M., & Levine, S. (2023). Offline RL for Natural Language Generation with Implicit Language Q Learning. https://arxiv.org/abs/2206.11871
>
> Chow, Y., Tulepbergenov, A., Nachum, O., Ryu, M., Ghavamzadeh, M., & Boutilier, C. (2022). A Mixture-of-Expert Approach to RL-based Dialogue Management. https://arxiv.org/abs/2206.00059

---

> > ### Comment · Reviewer_BhVQ · 2023-08-11
> > **Thanks for the rebuttal**
> >
> > **Diversity**
> >
> > Though I understand that your method promotes diversity, that doesn't mean you shouldn't also evaluate diversity if you want to claim it actually generates diverse dialogue. Just saying it probably will generate diverse conversations based on the method without evaluating it explicitly does not suffice.
> >
> > **Long-term planning**
> >
> > I still don't see how this experiment shows that your method is better at planning than just any non-bandit RL method. Again, the higher return might be due to other aspects.
> >
> > **Baseline comparison**
> >
> > I understand that you won't be able to surpass SotA LLMs without also using those as a base, of course, however my comment is directed at a baseline that helps in interpreting sentiment scores.
> >
> > **Human eval**
> >
> > Since you claim in the main text that your methods outperform offline RL baselinee, and since the human eval in Appendix E seems to show 2/3 of your methods do worse than baselines in terms of fluency and sentiment, actually showing the method is more sample-efficient seems important for the current work. I would like to see the human eval section brought to the main text.
> >
> > All in all, I remain with my points that the claims about the diversity and sample-efficiency as well as overall performance (shown by the human eval) are overstated, and will keep my rating.

---

> > > ### Author Response · Authors · 2023-08-11
> > >
> > > **Diversity**
> > >
> > > We appreciate your feedback on the evaluation of diversity. As previously mentioned, our work builds upon the work by Chow et al., 2023, which has already demonstrated that MoE dialogue managers are capable of generating diverse utterances with different experts. Similar to Chow et al., 2023, Tables 3 and 4 in our appendix also quantitatively showcase the diversity and skill-related scores of our MoE experts. Furthermore, our contribution, as illustrated in Figure 2, is to develop a compositional dialogue manager that better utilizes the distinctiveness of these experts while achieving higher returns. This implies that the diversity of our MoE utterances not only contributed to the different diversity of each MoE expert but also the utilization of diverse intents via RL. If deemed necessary, we can also add a diversity-based rater’s evaluation in the final paper.
> > >
> > >
> > > **Long-term planning**
> > >
> > > Thank you for raising this point. To the best of our knowledge, the standard measure in RL for evaluating long-term planning is via comparing cumulative returns. While we acknowledge that this might not be the perfect metric for dialogue planning, we emphasize that our primary objective in this paper is to introduce RL methods to the MoE framework, rather than to develop SOTA planning-based dialogue managers for particular applications. Our evaluation methodology is consistent with Jaques et al., 2020, which also utilized similar metrics and evaluations. That being said, designing evaluation methods to gauge the planning ability of dialogue managers presents an exciting avenue for future research. Please also let us know if you have any specific ideas in mind.
> > >
> > >
> > > **Baseline comparison**
> > >
> > > Our apologies for the earlier oversight. We understand now that you're suggesting using a Large Language Model (LLM) as an oracle for sentiment scoring. While it's a valuable suggestion and offers a compelling reward for future studies, in this work we decide to stick with a RoBERTa-based sentiment classifier for sentiment scoring because it is also what other related work, e.g., Jaques et al., 2020, used to set up their dialogue management environments. Nonetheless, when our methods are applied to larger-scale problems, adopting such an LLM score would be very beneficial. We will make a note of this in the final paper.
> > >
> > > **Human Evaluation**
> > >
> > > We acknowledge your concerns about the human evaluation results. It is important to highlight that most of our proposed methods (IQL, MoE-VRL, FtRL) significantly outperform earlier baselines, notably KLC and BC, in these open-domain dialogue management tasks, especially the offline RL methods we've designed for MoE (MoE-VRL, FtRL) also consistently outperform the KLC, BC baselines. We'd also like to mention, albeit cautiously, that human evaluations inherently have a degree of variability. However, we believe our results are indicative of the effectiveness of our approaches.
> > >
> > > **Sample Efficiency**
> > >
> > > Based on your feedback, we will revisit our claims regarding sample efficiency in the final paper. Our intention is to provide clarity and avoid overstating our results. In the revised manuscript, we will further temper our assertions to ensure accuracy and reduce any potential ambiguity.

---

> > > > ### Comment · Reviewer_BhVQ · 2023-08-14
> > > >
> > > > **Diversity**
> > > > I still think diversity should be explicitly evaluated with metrics that simply look at the diversity of the generated conversations, and as far as I can see, the paper by Jaques et al you mention does human evals on diversity as well. However, I see your point and upon re-reading the paper I see that the claim in the concluding remarks and abstract is specifically "diversity of intents" not diversity of utterances. I apologise for mixing the two up in my discussion above. I now agree with the authors that claims of diversity of intents are properly evaluated.
> > > >
> > > > **Long-term planning**
> > > > Regarding long-term planning; it seems to me that to claim this you need to use a benchmark for which a certain return can only be obtained if long-term planning is uses, is that the case here?
> > > >
> > > > As it stands, my rating remains a 6, which is an accepting and positive rating, based on the contributions and their evaluations as we discussed so far.

---

> > > > > ### Author Response · Authors · 2023-08-14
> > > > >
> > > > > We would first like to thank you for your follow-up responses, your feedback is greatly appreciated. We do admit that the proposed system has limitations and would try to incorporate the feedback you have provided in the final draft.
> > > > >
> > > > > **Long-Term Planning**
> > > > > You are correct – it is indeed a limitation inherent to automatic multi-turn dialogue evaluation. Currently,  there is a lack of benchmarks equipped with a clear reward function,  based on which we can conclude that obtaining a sufficiently high return would correspond to long-term planning. This in itself is an interesting problem, and very hard to solve, given the subjectivity of evaluation.
> > > > >
> > > > >
> > > > > Your proactive engagement and constructive feedback have been invaluable. We appreciate the time and effort you’ve invested in this review process and will certainly strive to incorporate your feedback into the final paper.

---

> > > > > > ### Comment · Reviewer_BhVQ · 2023-08-15
> > > > > >
> > > > > > Thank you for the responses and for bearing with me!

---

### Official Review · Reviewer_nRAM · 2023-07-06

**Soundness:** 3 good
**Presentation:** 2 fair
**Contribution:** 3 good
**Rating:** 4
**Confidence:** 4

**Summary:**

The authors of the paper proposed a suite of off-line reinforcement learning methods utilizing Mixture-of-Expert Language Models (MoE LMs) to train dialogue management agents. Moreover, they experimented their RL methods on two open-domain dialogue datasets and showed better overall performance of their methods (MoE specific offline RL) over SOTA offline RL methods.

**Strengths:**

S1: In their experimental section, it is shown that their MoE specific RL methods outperformed the SOTA offline RL methods.

S2: Human evaluation is also done by recruiting 80 workers.

S3: It is shown that those agents which have better performance utilize all the knowledge of different experts in a balanced way.

**Weaknesses:**

W1: In the conclusion section of the paper, it is stated that their specialized offline RL methods have better sample efficiency, however, I did not see any experimental proof for it.

W2: In the introduction section (page 2), they described their first component of their methods two times: “Our methods consist of three main components: 1) a primitive LM which, using a probabilistic encoder and decoder, is capable of generating diverse semantic intents 1) a primitive LM that uses a probabilistic encoder-decoder pair to generate sentences with diverse semantics and intents”.

W3: In the section 6 (page 7) before experiment 1, the second appendix is not referenced properly: “More details and results can be found in Appendix E and ??”

W4: The implementation is not provided so that one cannot reproduce their results.

**Questions:**

See weakness section.

---

> ### Author Rebuttal · Authors · 2023-08-09
>
> We thank the reviewers for the useful feedback aimed at improving our paper.  Please find individual responses to comments below.
>
> ### Conclusion indicates improved sample efficiency
> The improved efficiency arises from both modeling and algorithmic choices. First,  we adopt the MoE LM framework (Chow et al., 2023) and greatly simplify the action space of the RL dialogue management problem, as we no longer rely on RL to directly control the token-level autoregressive generation of the language model but rather to select the best utterance to output at the current conversation turn from the pool of candidate responses generated by MoE-based model. Second, we developed our offline dialogue RL planning algorithms for MoE-LMs under the IQL offline RL methodology, which has been shown in Snell et al., 2023  to have improved performance. While these factors intuitively improve sample efficiency, we acknowledge the reviewer’s comment that explicit analysis has not been conducted in our paper, and therefore we will update the conclusion section in the final paper to clarify these points and soften our claims about sample efficiency improvement.
>
>
> ### Typo in the introduction section
> Thanks for catching this, we will remove the duplicate texts in the final draft.
>
> ### Reference in the experiment section
> We apologize for the formatting issue, we meant to say Appendix A and E, we will fix the appendix references in the final version of the paper.
>
> ### Implementation details are unclear
> Unfortunately due to IP concerns at this point, our institution has not approved our request for open-sourcing code. We indicated that restriction in our initial submission checklist and will try to release the code by the final submission timeline. In the meantime, we tried our best to provide detailed explanations about experimental setup, model architectures, and RL training procedures in Appendix B to D (and we also follow the implementation details of the original MoE paper: Chow et al., 2023), so that the reader can implement these concepts.
>
> ### References
>
> Snell, C., Kostrikov, I., Su, Y., Yang, M., & Levine, S. (2023). Offline RL for Natural Language Generation with Implicit Language Q Learning (arXiv:2206.11871). arXiv. https://arxiv.org/abs/2206.11871
>
> Chow, Y., Tulepbergenov, A., Nachum, O., Ryu, M., Ghavamzadeh, M., & Boutilier, C. (2022). A Mixture-of-Expert Approach to RL-based Dialogue Management (arXiv:2206.00059). arXiv. https://arxiv.org/abs/2206.00059

---

### Official Review · Reviewer_3gha · 2023-07-07

**Soundness:** 3 good
**Presentation:** 2 fair
**Contribution:** 3 good
**Rating:** 6
**Confidence:** 2

**Summary:**

This paper introduces multiple reinforcement learning algorithms for dialogue management, in particular when combined with mixture-of-expert language models. Generally, a primitive (general) language model, as well as expert language models which have a specific intent or personality, generate candidate utterances. The dialogue management module then learns to choose among them.

**Strengths:**

The paper introduces multiple novel RL algorithms tailored for dialogue management.

Using RL-based dialogue management over mixture-of-expert LMs is well-motivated. The language models may generate fluent and diverse outputs, while the limited size of the action space allows for efficient learning.

Coordinating multiple language models (or especially a single language model with different adaptors/parameter-efficient fine-tuning modules) could be impactful by improving generation quality and diversity given a fixed number of parameters.

**Weaknesses:**

The description of the algorithms is quite dense. As multiple approaches are introduced, a more thorough discussion of which one(s) to prefer under certain conditions would be helpful.

The number of turns is fixed in the experiments. Conversations could be much richer if the conversation length was more flexible.

Without sharing the code, some of the experiments may be difficult to reproduce.

The results in tables 1 and 2 are difficult to interpret.



**Questions:**

Could you clarify "Our experiments demonstrate that model-based evaluation can significantly improve dialogue management over the model-free counterpart [...]"? Does the evaluation method change the DM policy?

Could you describe the evaluation approaches in more detail? What do the reported numbers exactly represent?

Could the approach be adapted to work without a primitive LM (i.e. only expert LMs with a specific personality)?

Do you have examples of conversations generated with different approaches?

[L49/50] Component #1 is repeated.

**Limitations:**

Some limitations are discussed in the appendix.

---

> ### Author Rebuttal · Authors · 2023-08-09
>
> We thank the reviewers for the useful feedback and appreciation of the prospect of our work. Please find individual responses to comments below.
>
> ### Dense algorithmic descriptions
> Our contributions are on developing several MoE-specific offline RL algorithms and comparing different offline RL approaches on MoE dialogue management, which results in the sheer amount of technical details in the main paper. We will shift some of these details to the appendix and add more intuitive explanations of different methods (when one is preferred over another in practical scenarios) in the final paper.
>
> ### Fixed conversation turns in experiments
> In our experiments, the conversation turn has been fixed as a way to evaluate the different methods. This quantity can be regarded as the horizon of the RL problem. The underlying number of conversation turns can still vary because the user/agent can indicate an end of the conversation by outputting an EOS token before the maximum conversation turn is reached. To check for the quality/richness of our generated conversations, we also evaluate our methods in raters studies whose results are given in Appendix E.
>
> ### Tables 1 and 2
> We acknowledge the reviewer’s confusion caused by the condensed presentation of these tables and will include more detailed explanations in the final paper. These tables present the average return (discounted sum of per-turn rewards in the dialogue conversations) of the dialogue agent of interest, accumulated over a 5-turn conversation. The return is averaged over 100 conversations, and the standard error is also provided.  A higher value indicates the corresponding agent is able to perform better dialogue planning and results in better overall sentiment improvement.
>
>
> Table 1 and Table 2 respectively compare the results between offline RL methods (that are not necessarily adapted to the MoE framework) and offline RL methods that are specifically designed with the MoE framework in mind. We can see that MoE-specific offline RL methods can manage to perform significantly better than standard offline RL methods in our dialogue management experiments.
>
> ### Model-based/free evaluation
> Model-based and model-free approaches differ in the way of policy extraction. In the case of model-free methods the MoE policy only relies on the learned Q-function as the scoring function, while in the model-based experiments the policy is constructed as the softmax the value-to-go function (V function) that corresponds to to the augmented conversation history w.r.t. the predicted next-user utterance (given by the utterance prediction model that takes conversation history and candidate utterances as inputs)
>
> Details of evaluation approach: Each numerical result in our evaluation is the averaged return of 100 conversations driven by the RL agents of interests. As mentioned in Line 328, this metric corresponds to the discounted sum of the reward for each dialogue turn. The reward function is defined on the user sentiment score in the next conversation turn (affected by current conversation history and the bot response) as described in Line 318.
>
> ### Without primitive LM
> (Please let us know if we mis-interpret your question) Yes, while our offline RL approach is built on top of the more general MoE framework that consists of a universal representation space that embeds diverse semantics and a gamut of various expert generators, it does not rely on the details of these experts. Therefore, our approach can also be directly applied to a (simpler) setting when there is only an expert LM with a particular personality.
>
> ### Code sharing
> Unfortunately due to IP concerns at this point our institution has not approved our request for open-sourcing code. We already indicated that restriction in our initial submission checklist and will try to release the code by the final submission timeline. In the meantime, we tried our best to provide detailed explanations about experimental setup, model architectures, and RL training procedures in Appendix B to D (and we also follow the same MoE implementation details illustrated in the original paper: Chow et al., 2023 ) so that the reader can implement these concepts.
>
> ### Sample conversations
> We apologize for not including the sample conversation, we have included some sample conversation in the extra PDF posted above and will include that in the final paper.
>
> ### Typos
>
> Thanks for catching this, we will correct it in the final draft.
>
> ### References
> Chow, Y., Tulepbergenov, A., Nachum, O., Ryu, M., Ghavamzadeh, M., & Boutilier, C. (2022). A Mixture-of-Expert Approach to RL-based Dialogue Management (arXiv:2206.00059). arXiv. https://arxiv.org/abs/2206.00059

---

> > ### Comment · Reviewer_3gha · 2023-08-21
> >
> > Thank you for your response.
> >
> > For `Without primitive LM`, I wanted to more clearly understand the differences between $\mathcal{G}_{\ge 1}$ and $\mathcal{G}_0$. $\mathcal{G}_0$ appears to have a particular and distinct status, but the notation is very similar to the expert distributions.
> >
> > Could it make any sense to only have $\mathcal{G}_{\ge 1}$, but not $\mathcal{G}_0$?

---

> > > ### Author Response · Authors · 2023-08-21
> > >
> > > Thank you for your comment.
> > >
> > > We will try to clarify the difference between  $\mathcal{G}_{\ge 1}$ and $\mathcal{G}_0$.
> > >
> > > $\mathcal{G}\_0$ has a distinct status in methodology, where it was designed and trained to discover the semantic space via an embedding (and a generic sampler for sampling in this latent embedding space), while also learning the encoder and decoder. On the other hand, $\mathcal{G}\_{\ge 1}$ leverages the encoder and decoder learned by $\mathcal{G}\_{0}$,  but further finetunes the latent space sampler to represent the respective experts' intent/behavior, using the sentiment based reward. Yet, when integrated into RL method for DM, both entities operate *without* any distinction.
> > >
> > > To address your query, you are right, it would be perfectly fine to only have $\mathcal{G}_{\ge 1}$, but not $\mathcal{G}_0$.

---

### Official Review · Reviewer_T4hv · 2023-07-07

**Soundness:** 3 good
**Presentation:** 2 fair
**Contribution:** 3 good
**Rating:** 6
**Confidence:** 2

**Summary:**

The authors address offline RL training of dialogue models. They note
that gathering trajectories with humans-in-the-loop is potentially
expensive (and perhaps even dangerous). Their method is a MoE
involving a general LM and several specialized intent-accounting-for
LMs, which generate candidates for a dialogue manager to select from:
this process reduces the action space significantly (because selecting
the next utterance becomes a task at the utterance level, rather than
at the word level). The reward is a roberta sentiment classifier
applied to simulated dialogue responses. Experiments compared to
several single-policy/MoE methods show the proposed method is better
able to elicit positive sentiment responses in simulation.

**Strengths:**

The authors study an interesting problem --- offline RL for dialogue,
and frame the task well --- DialoGPT offers a nice simulation
environment, and I understand the motivation for the reward function
the authors optimize. I commend the authors for working in a
multidisciplinary area, mixing RL and dialogue modeling contributions.
The experiments cover a broad range of RL algorithms, as well as 2
dialogue corpora.

**Weaknesses:**

- I would have liked to have seen some human evaluations of response
  quality/elicited sentiment (I think the question that
  operationalizes the reward may be: "Which model's response would
  cause you to write a more positive response?"). Current evaluations
  are limited to reward optimization, which makes sense from an RL
  perspective. But, at the very least (if a small-scale human
  evaluation were prohibitive), it was odd to see a paper about
  dialogue with no dialogues shown.

- The reward seems a bit reductive --- 1) What is the RoBERTa
  classifier trained on? Unfortunately, Liao et al. 2021 is locked
  behind a paywall so I can't check. I know sentiment models are often
  trained on specific domains (e.g., movie reviews or yelp reviews),
  so it might not generalize well (I am fearful of reward hacking);
  and 2) should it really be the case that the optimized reward be
  entirely based on sentiment? This seems like it might risk just
  pushing the dialogue manager to be a sycophantic "yes-man". I would
  have appreciated some justification for this choice. (in fact, few
  potential limitations of the work are discussed --- this type of
  reflection would have been appreciated).

- The model itself seems a bit convoluted --- the idea of compressing
  the primitive encoder's output into a single vector (and then
  sampling from a mixture of normal distributions conditioned on that
  encoding as the MoE), felt a bit roundabout. Why not, e.g., not
  pool, and train a separate decoder for each expert, ala T5? Latent
  variable modeling is cool, but it felt superfluous, even to the core
  message of this paper (which, for me, focuses on MoE for state space
  reduction) --- I at least would have liked to have seen the ablation
  of simpler methods, e.g., over-generation with differently prompted
  LMs in a zero-shot way.


UPDATE:

the authors do have human eval (and will surface from the appendix) and made a few more clarifications in response). I have raised my score.

**Questions:**

- What is the reward model from Liao et al. 2021 trained on?
- Is it possible to run human evaluation or include discussion of the outputs to make sure reward hacking isn't occurring?
- Why the latent variable model instead of, e.g., just encoder/decoder? It seems a bit orthogonal, and some simpler baselines suggested by this encoder/decoder choice are missing.
- Can better discussion be added for potential limitations of this particular reward in this particular setup?

**Limitations:**

No, see above.

---

> ### Author Rebuttal · Authors · 2023-08-09
>
> We thank the reviewer for their useful feedback. Please find individual responses to comments below:
>
> ### Human evaluation
> We conducted a human evaluation of the generated conversations by asking raters to score how our MoE bot managed to improve the overall sentiment within the conversations with different RL algorithms. The results can be found in  Appendix E (Table 10) which aims to compare the overall fluency and sentiment improvement of the conversation. Details of the rater evaluation can be found in the Appendix. This experiment sheds light on how our chosen automatic evaluation metrics align with human evaluations on demonstrating how our offline RL methods, when paired with MoE models, can improve the overall conversation objective. We will also move these experimental results into the main paper (given the extra page provided), as they are important to demonstrate the advantages of using specialized offline RL methods. Additionally, Appendix A2 contains evaluations of the MoE embedding space and different experts (Tables 3 and 4), echoing the original MoE paper by Chow et al., 2023.
>
> ### Sample dialogues:
> We acknowledge the reviewer’s comments on the lack of sample dialogues. We focused on displaying quantitative studies in the original submission. We included a snippet of sample dialogues in our rebuttals to showcase the effectiveness of our methods in sequential conversations, and we will add more sample dialogues in Appendix E in the final version.
>
> ### Sentiment Classification Model
> We regret the confusion regarding the referencing of sentiment classifiers. In our work, we utilized the HuggingFace RoBERTa model trained on the Twitter dataset to recognize sentiment (https://huggingface.co/cardiffnlp/twitter-roberta-base-sentiment-latest). Initially, we cited Liao et al., 2021, aiming for specificity in sentiment classification models.
>
> Acknowledging the reviewer's concern about potential confusion, we will update the reference to TimesLM (Loureiro et al., 2022), an open-source model utilized beyond sentiment analysis. It's essential to clarify that our paper's primary focus is on introducing specialized offline RL methods for dialogue management within the mixture of expert frameworks. The selection of the sentiment classifier and reward design are experimental decisions, unrelated to the underlying RL algorithms.
>
> ###  Reward Hacking and Model Generalization:
> We acknowledge the reviewer’s concern about reward hacking in sentiment optimization. As mentioned above, our experiments are meant for demonstrating the effectiveness of different offline RL dialogue management methods rather than developing a full-blown universal conversation bot. We decided to optimize w.r.t. user sentiment transition partially because of Table 2b of Jaques et al., 2020, which experimented with the same 2 conversation datasets and showed that the user sentiment signal is the most correlated with real human feedback of conversation quality (measured w.r.t. rater’s upvotes).
>
> To avoid language model overfitting, our agent adopted the MoE framework that restricts dialogue responses to be selected from among the set of dialogue utterances generated by experts of different skills (e.g., positive/negative sentiment, semantic coherence/diversity, question, etc.). Experiment results show that our RL method tends to select a more diverse set of experts (Figure 2a) to avoid sycophantic responses. Human evaluation (Appendix E) further shows that our agent tends to be more fluent and leads to user sentiment improvement over multiple turns.
>
> ### MoE framework; Shared latent is superfluous; Separate decoder per expert
>
> The MoE framework's core idea lies in representing multiple experts within a language model across various parts of the semantic latent space. This allows the experts to generate candidate utterances in a modular way that is suitable for downstream dialogue management tasks. This approach not only lessens computational demands by enabling a range of responses with different intents but also streamlines the token-level MDP formulation in dialogue RL, resulting in more effective management. For a detailed motivation, refer to the original work by Chow et al., 2023. Our research builds on this, focusing on enhancing offline RL capabilities, and we'll enrich the paper's introduction with more insights into MoE-LM.
>
> The shared encoder in the MoE framework, trained for both accuracy and diversity, encodes the conversation history into a versatile embedding space. This serves as the foundation for expert utterance generation (phase 2) and MoE-MDP state space (phase 3). Specifically, it permits (i) the creation of responses with varied intents by sampling from specific latent space regions, and (ii) RL planning in a reduced, continuous state space (see Eqn 4-6 in our paper). Utilizing the same decoder minimizes distribution shifts during data generation, simplifying RL training compared to non-MoE approaches (refer to the original paper's Table 2 for details and ablation studies).
>
> ### Better reward discussions
> Our RL reward choices follow primarily from the open-domain offline RL dialogue management paper: Jaques et al., 2020, and the MoE paper: Chow et al., 2023. Both for fluency/coherence (quality) and sentiment improvement (task success). We will add additional discussions regarding RL reward choices in the final paper.
>
> ### References
> Chow, Y., Tulepbergenov, A., Nachum, O., Ryu, M., Ghavamzadeh, M., & Boutilier, C. (2022). A Mixture-of-Expert Approach to RL-based Dialogue Management. https://arxiv.org/abs/2206.00059
>
> Jaques, N., Shen, J. H., Ghandeharioun, A., Ferguson, C., Lapedriza, A., Jones, N., Gu, S. S., & Picard, R. (2020). Human-centric Dialog Training via Offline Reinforcement Learning. http://arxiv.org/abs/2010.05848
>
> Loureiro, D., Barbieri, F., Neves, L., Anke, L. E., & Camacho-Collados, J. (2022). TimeLMs: Diachronic Language Models from Twitter. https://arxiv.org/abs/2202.03829

---

> > ### Comment · Reviewer_T4hv · 2023-08-15
> > **Thanks!**
> >
> > Thanks for your response. Here are some response-responses :
> >
> > - Human eval: thank you, this is great! I did not see these and would definitely recommend moving them into the main body.
> > - Thanks for sharing these! I think, ideally, in addition to showing a few examples, some commentary could be added about the relative strengths of the predictions of each approach. Even better would be an error analysis. But --- thanks for this, it helps!
> > - Sentiment: thanks for the clarification
> > - MoE vs. a simpler baseline. I do appreciate the authors points about the advantages of MoE. But I do feel that training a separate decoder for each intent/expert is a similarly simple-to-train baseline that would have been nice to see.
> >
> > I will raise my score in light of these updates.

---

### Author Rebuttal · Authors · 2023-08-09

## Sample Utterances

We acknowledge the reviewer’s comments on the lack of sample dialogues. We focused on displaying quantitative studies in the original submission. We included a snippet of sample dialogues in the attached PDF to showcase the effectiveness of our methods in sequential conversations, and we will add more sample dialogues in Appendix E in the final version.

---

### Decision · Program_Chairs · 2023-09-21

**Decision:**

Accept (poster)

**Comment:**

This paper proposes off-line RL methods using Mixture-of-Expert (MoE) LM (involving a general LM and several specialized intent-accounting-for LMs) to train dialogue policy and outperform SOTA offline RL, showing that those agents can utilize all the knowledge from different experts in a balanced way.

This paper provides a reasonable and more flexible solution for a dialogue manager to learn its policy. By leveraging the capability of LMs, this approach can allow a dialogue agent to be better controlled and also take advantage of powerful LLMs (e.g. ChatGPT; end-to-end conversational models). This direction may attract a lot of interest because leveraging diverse powerful LLMs may be a practical solution for building commercial products.